# Intersection of TKS5 and FGD1/CDC42 signaling cascades directs the formation of invadopodia

Anna Zagryazhskaya-Masson[1]*, Pedro Monteiro[1]*, Anne-Sophie Macé[1,2], Alessia Castagnino[1], Robin Ferrari[1], Elvira Infante[1] , Aléria Duperray-Susini[1], Florent Dingli[3] , Arpad Lanyi[4], Damarys Loew[3] , Elisabeth Génot[5,6] , and Philippe Chavrier[1]

**Tumor cells exposed to a physiological matrix of type I collagen fibers form elongated collagenolytic invadopodia, which differ from dotty-like invadopodia forming on the gelatin substratum model. The related scaffold proteins, TKS5 and TKS4, are key components of the mechanism of invadopodia assembly. The molecular events through which TKS proteins direct collagenolytic invadopodia formation are poorly defined. Using coimmunoprecipitation experiments, identification of bound proteins by mass spectrometry, and in vitro pull-down experiments, we found an interaction between TKS5 and FGD1, a guanine nucleotide exchange factor for the Rho-GTPase CDC42, which is known for its role in the assembly of invadopodial actin core structure. A novel cell polarity network is uncovered comprising TKS5, FGD1, and CDC42, directing invadopodia formation and the polarization of MT1-MMP recycling compartments, required for invadopodia activity and invasion in a 3D collagen matrix. Additionally, our data unveil distinct signaling pathways involved in collagenolytic invadopodia formation downstream of TKS4 or TKS5 in breast cancer cells.**

## Introduction

Migration of cells through tissues is essential during embryonic development, tissue repair, and immune surveillance (Madsen and Sahai, 2010). Deregulated invasive migration is also a key event in diseases, including cancer dissemination. Because of a high degree of intra- and intermolecular covalent cross-links in type I collagen in native tissues that prevent physical expansion of preexisting ECM pores and cell invasion, proteolytic degradation is indispensable for ECM penetration by cancer cells (Rowe and Weiss, 2008; Sabeh et al., 2009; Wolf et al., 2013). Several studies based on both in vitro and in vivo assays revealed that invasive cancer cells negotiate tissue barriers by forming specialized F-actin–based protrusions called invadopodia, which focally degrade the ECM, enabling cell penetration (Castro-Castro et al., 2016; Gligorijevic et al., 2014; Leong et al., 2014; Linder et al., 2011; Murphy and Courtneidge, 2011). MT1-MMP, a trans-membrane matrix metalloproteinase, is concentrated at invadopodia and is essential for pericellular matrix degradation and carcinoma cell invasion across the basement membrane and dense collagen tissues (Feinberg et al., 2018; Hotary et al., 2006; Lodillinsky et al., 2016; Perentes et al., 2011; Wolf et al., 2007).

Studies using tumor cells plated on a thin layer of gelatin revealed that invadopodia formation is a multistep process initiated by the assembly of F-actin and cortactin-positive invadopodia precursors (Eddy et al., 2017). Precursors are progressively stabilized and gain matrix degradative capacity as MT1-MMP accumulates during invadopodia maturation (Artym et al., 2006; Branch et al., 2012; Eddy et al., 2017; Mader et al., 2011; Oser et al., 2009; Sharma et al., 2013). Although the complete sequence of events involved in invadopodia maturation is missing, recruitment of the scaffold protein TKS5 (tyrosine kinase substrate with five SH3 domains; aka SH3PXD2A, FISH) is a key step for the maturation of short-lived actin-based precursors into matrix degradation–competent invadopodia (Eddy et al., 2017; Sharma et al., 2013). The related scaffold proteins, TKS4 (aka SH3PXD2B) and TKS5, have been identified as c-Src substrates and as critical regulators of invadopodia and podosome formation and function (Buschman et al., 2009; Dülk et al., 2018; Seals et al., 2005). In addition, several studies have highlighted key roles for TKS4 and TKS5 proteins in tumor growth and metastasis in vivo (Blouw et al., 2015; Eckert et al., 2011; Iizuka et al., 2016;

.............................................................................................................................................................................................................................

[1]Institut Curie, PSL Research University, Centre National de la Recherche Scientifique, UMR 144, Paris, France; [2]Cell and Tissue Imaging Facility (PICT-IBiSA), Institut Curie, PSL Research University, Centre National de la Recherche Scientifique, Paris, France; [3]Mass Spectrometry and Proteomic Laboratory, Institut Curie, PSL Research University, Paris, France; [4]Department of Immunology, Faculty of Medicine, University of Debrecen, Debrecen, Hungary; [5]European Institute of Chemistry and Biology, Bordeaux, France; [6]Centre de Recherche Cardio-Thoracique de Bordeaux, Institut National de la Santé et de la Recherche Médicale U1045, and Université de Bordeaux, Bordeaux, France.

*A. Zagryazhskaya-Masson and P. Monteiro contributed equally to this paper; Correspondence to Pedro Monteiro: pedro.monteiro@curie.fr; Philippe Chavrier: philippe.chavrier@curie.fr.

Leong et al., 2014). TKS proteins harbor four (TKS4) to five (TKS5) SH3 domains involved in interactions with P-rich motifs on partner proteins, and a phox homology (PX) domain that binds the plasma membrane phosphoinositide, phosphatidylinositol-bisphosphate (PI(3,4)P$_2$; Abram et al., 2003; Buschman et al., 2009; Lányi et al., 2011; Saini and Courtneidge, 2018). TKS5 interacts with N-WASP (neuronal Wiskott–Aldrich syndrome protein) through its SH3 domains, and c-Src–phosphorylated TKS5 interacts with Nck, linking TKS5 to invadopodial F-actin assembly and ECM degradation (Oikawa et al., 2008; Seals et al., 2005; Stylli et al., 2009). It remains to be established whether TKS5 (and TKS4) may be linked to other key invadopodia regulatory signaling modules, such as the CDC42 pathway, which plays a central role in invadopodial actin assembly and invadopodia function (Ayala et al., 2009; Chander et al., 2013; Di Martino et al., 2014; Pichot et al., 2010; Sakurai-Yageta et al., 2008; Yamaguchi et al., 2005; Yamamoto et al., 2011).

Recent work revealed that invadopodia structure and activity differ depending on the composition and mechanical properties of the matrix environment (Artym et al., 2015; Juin et al., 2012; Parekh et al., 2011). In the classic model used to study invadopodia formation, cancer cells are plated on a thin (quasi-2D) substratum of denatured collagen (i.e., gelatin) where degradative activity is concentrated in 0.1–0.5-µm-diameter, actin-rich puncta (Linder et al., 2011). By contrast, when exposed to more physiological matrix construct of type I collagen fibers, cancer cells assemble cortactin- and F-actin–positive, collagenolytic invadopodia in association with the collagen fibers, which can be several micrometers in length (Castagnino et al., 2018; Infante et al., 2018; Juin et al., 2012; Monteiro et al., 2013). Whether and how the related TKS4 and TKS5 proteins play a role in the assembly of collagenolytic invadopodia is currently unknown. Here, we set out to characterize the contribution and mechanisms of action of TKS proteins in the formation of collagenolytic invadopodia in breast cancer cells. Using a proteomic approach, we immunoprecipitated and identified a set of TKS5 interactors, including faciogenital dysplasia protein 1 (FGD1), a highly specific CDC42 guanine exchange factor (CDC42-GEF; Zheng et al., 1996). We validated the interaction between FGD1 and TKS5 by coimmunoprecipitation and GST pull-down assays and further mapped their interacting domains. We found that TKS5 and FGD1 colocalize in collagenolytic invadopodia and are required for invadopodia formation and activity in invasive MDA-MB-231 triple-negative breast cancer (TNBC) cells and HT-1080 fibrosarcoma cells. We further identified a signaling module comprising TKS5 and FGD1 and its target Rho GTPase, CDC42, in the polarization of MT1-MMP storage compartments required for collagenolytic invadopodia activity and 3D collagen invasion. In addition, in Hs578T TNBC cells, we found that TKS4 localized to collagenolytic invadopodia, but in contrast to TKS5, TKS4 does not interact with FGD1 and operates through a CDC42-independent mechanism. Our data unveil distinct signaling pathways involved in collagenolytic invadopodia formation and cell polarity downstream of the scaffolding TKS4 and TKS5 proteins in cancer cells.

## Results

### TKS5 is required for the formation of collagenolytic invadopodia

Invasive MDA-MB-231 breast tumor cells cultured on a layer of fibrillar type I collagen for 60–90 min formed curve-shaped structures, up to several micrometers in length, in association with the underlying fibers (Fig. 1 A). These structures were enriched in cytoskeletal components including F-actin, cortactin, and the invadopodia scaffolding protein, TKS5 (Fig. 1 A and Fig. S1 A). Staining with Col1-3/4C antibody that detects the collagenase-cleaved fragment of collagen I showed a robust collagenolytic activity associated with TKS5$^{GFP}$-positive structures (Fig. S1 B). Correlation of TKS5 and cortactin pixel fluorescence intensity along elongated invadopodia based on linescan analysis revealed a strong association of the two markers (Fig. 1 B). In addition, the collagenolytic activity of MDA-MB-231 cells strongly increased upon TKS5$^{GFP}$ overexpression (Fig. 1, C and D). In contrast, TKS5 silencing impaired invadopodia formation and led to an ~70% inhibition of collagen cleavage by MDA-MB-231 cells, comparable to the loss of MT1-MMP itself (Fig. S1, C–E; and Fig. S2, A and B). Decreased collagenolysis activity correlated with a drastic ~75% reduction of the capacity of TKS5-silenced cells to invade through a 3D collagen gel, similar to the effect of MT1-MMP knockdown (Fig. 1, E and F). Altogether, these observations confirmed that elongated TKS5-positive structures were bona fide collagenolytic invadopodia consistent with the strong proinvasive and prometastatic potential of TKS5 (Ferrari et al., 2019b; Juin et al., 2012; Monteiro et al., 2013; Saini and Courtneidge, 2018).

### FGD1 is a novel TKS5 partner in invadopodia

To characterize TKS5 downstream signaling pathways involved in invadopodia function, we generated MDA-MB-231 cells stably expressing TKS5$^{GFP}$, and lysates were prepared from cells cultured for 90 min on a layer of fibrillar type I collagen to induce a robust invadopodial response. A biological triplicate of anti-GFP immunoprecipitation followed by tandem mass spectrometry (MS) analysis was performed using MDA-MB-231 cells that did not express TKS5$^{GFP}$ as a negative control. Several proteins that were specifically identified in the immunoprecipitates from TKS5$^{GFP}$-expressing cells had been already identified as TKS5 partners, some with known invadopodia localization and/or function, validating our approach (Table S1). Among the potential TKS5 partners identified by this approach, FGD1, a CDC42-specific GEF (Zheng et al., 1996), was an interesting hit, as it is known to be required for podosome and invadopodia function using the gelatin substratum model (Ayala et al., 2009; Daubon et al., 2011). Of note, these findings confirmed a previously identified TKS5/FGD1 interaction by a global human interactome study (Hein et al., 2015).

We initially confirmed the interaction of TKS5$^{GFP}$ with endogenous FGD1 by immunoprecipitation followed by immunoblotting analysis (Fig. 2 A). Reciprocally, FGD1$^{GFP}$ could be coimmunoprecipitated with endogenous TKS5 (Fig. 2 B). A GST pull-down assay was used to narrow down the TKS5-binding domain of FGD1 to its amino-terminal P-rich domain (PRD; Fig. 2 C). Finally, we found that the PRD domain of FGD1 pulled

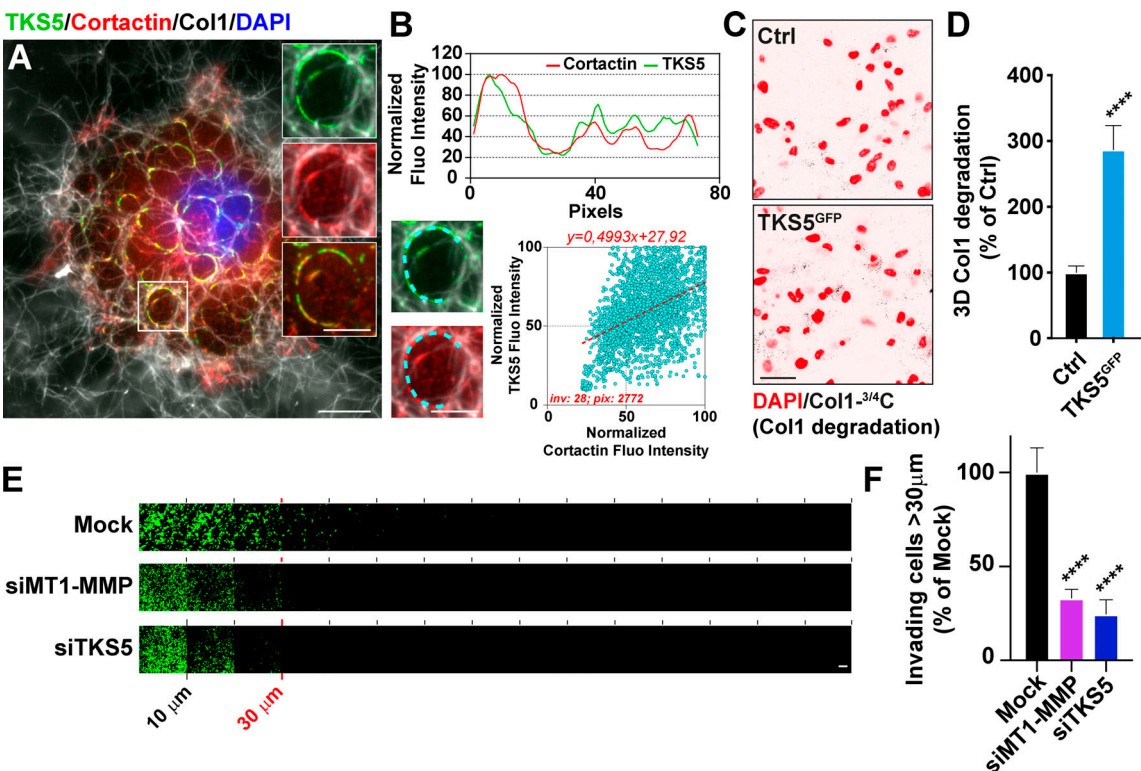

Figure 1.    **TKS5 is required for collagenolysis and for 3D collagen invasion by breast cancer cells. (A)** MDA-MB-231 cells were cultured on a fibrillary layer of type I collagen (gray) for 60 min and stained for cortactin (red), TKS5 (green), and DAPI (blue). Scale bar, 10 μm; zoom-in of boxed region, 5 μm. **(B)** Upper panel: Fluorescence (Fluo) intensity profiles for cortactin and TKS5 were recorded along the invadopodium (cyan dashed line in the insets) and normalized to the maximum fluorescence intensity set to 100. Lower panel: Correlation of cortactin (x axis) and TKS5 (y axis) pixel fluorescence intensity along 28 invadopodial structures. inv, number of invadopodia analyzed; pix, total number of pixels analyzed for both markers. **(C)** Representative images of pericellular collagenolysis detected with Col1-3/4C antibody (black signal in the inverted images). Nuclei were stained with DAPI (red). Scale bar, 50 μm. **(D)** Pericellular collagenolysis by the indicated MDA-MB-231 cell populations measured as mean intensity of Col1-3/4C signal per cell ± SEM. Values for control cells were set to 100%. Mann–Whitney U tests. **(E)** siRNA-treated cells were allowed to invade type I collagen plugs in an inverted invasion assay. After 48 h of invasion, nuclei were stained with DAPI (green), and serial optical sections (10-μm interval) were acquired. Scale bar, 100 μm. **(F)** Relative invasion of cells penetrating 3D collagen to depths ≥30 μm (see E). Data represent mean ± SEM normalized to invasion of mock-treated cells from three independent experiments. Kruskal–Wallis test. ****, P < 0.0001.

down a TKS5 truncated construct encompassing the fourth and fifth carboxy-terminal SH3 domains of TKS5 (Fig. 2 D).

Immunofluorescence analysis revealed extensive colocalization of TKS5GFP or FGD1GFP with their endogenous FGD1 or TKS5 partner in invadopodia, respectively (Fig. 3, A and B). In addition, FGD1 colocalized with the invadopodial protein marker, cortactin (Fig. 3 C). Live-cell imaging confirmed the strong colocalization of overexpressed TKS5 and FGD1 proteins in numerous highly dynamic invadopodial structures forming in association with the underlying collagen fibril network (Fig. 3 D and Video 1). TKS5- and FGD1-positive invadopodia had a strong capacity to remodel and clear the underlying collagen fibrils away from the cell body (Video 1). The expression of several key invadopodia components was analyzed in MDA-MB-231 cells in comparison with two additional invasive cell lines, Hs578T TNBC cells and HT-1080 fibrosarcoma cells. Immunoblotting analysis revealed similar expression profiles for the invadopodia markers in MDA-MB-231 and HT-1080 cell lines, whereas Hs578T cells expressed two lower molecular weight TKS5 isoforms (Fig. 3 E and see below). Similar to MDA-MB-231 cells, collagenolytic invadopodia enriched in cortactin, TKS5, and

FGD1 were observed in association with the collagen fibers in HT-1080 fibrosarcoma cells (Fig. 3, F–H). Collectively, our findings identified FGD1 as a novel interacting partner of TKS5 in matrix-degradative invadopodia in different cancer cell lines.

## Interaction of FGD1 with TKS5 is required for collagenolytic invadopodia formation and function

We went on investigating whether FGD1 contributed to the assembly and/or function of collagenolytic invadopodia in MDA-MB-231 cells. FGD1 knockdown (Fig. S2 C), induced a partial, but significant, 35–50% reduction of TKS5-positive invadopodia assembly (Fig. 4, A and I; and Fig. S3 A). Using identical knockdown conditions, reduction of FGD1 levels strongly inhibited the collagenolytic activity of MDA-MB-231 cells, similar to TKS5 knockdown (Fig. 4 B and Fig. S3, B and C).

As shown above, the PRD domain of FGD1 interacts with the carboxy-terminal region of TKS5 containing two SH3 domains known to interact with P-rich motifs (Fig. 2). Therefore, a W-to-A mutation in a conserved W residue known to be critical for binding to P-rich motifs was introduced in the SH3#4 (W861A) and SH3#5 (W1092A) domains of TKS5 (Fig. 4 C; Tanaka et al.,

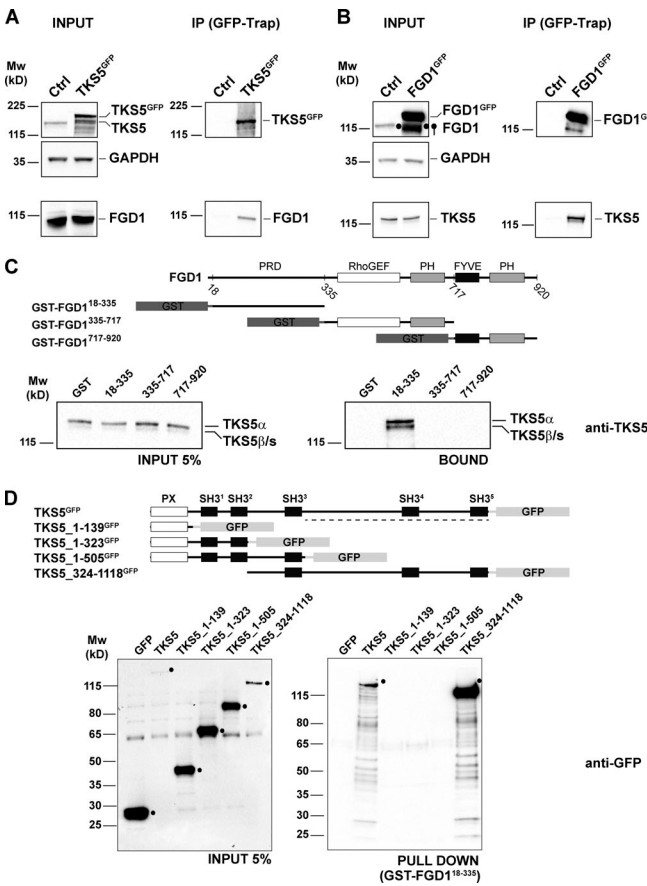

**Figure 2. The carboxy-terminal SH3 domains of TKS5 interact with the PRD domain of FGD1. (A)** Lysates of MDA-MB-231 cells expressing TKS5[GFP] or not (Ctrl) were immunoprecipitated with GFP antibodies. Bound proteins were analyzed with TKS5 and FGD1 antibodies (immunoprecipitation, IP). 5% of total lysate was loaded as a control (input). Equal loading was controlled using GAPDH antibody. **(B)** Lysates of MDA-MB-231 cells expressing FGD1[GFP] or not (Ctrl) were analyzed by immunoprecipitation as in A. **(C)** GST or indicated GST-FGD1 constructs on beads were incubated with MDA-MB-231 cell lysates. TKS5 protein was identified by immunoblotting analysis with TKS5 antibody. 5% of total lysates was loaded as a control (input). A schematic representation of FGD1 and GST constructs is shown. The endogenous TKS5 protein pulled down by GST-FGD1[18–335] represents ∼1.5% of the input. **(D)** Pull-down assays of indicated TKS5[GFP] constructs from MDA-MB-231 cell lysates with GST-FGD1[18–335] protein immobilized on beads. Bound material was analyzed with GFP antibody. 5% of total lysates was loaded as a control (input). A schematic representation of the TKS5 constructs is shown on the upper part. The GFP-tagged TKS5 and TKS5_324-1118 proteins pulled down by GST-FGD1[18–335] represent 1.35% and 2.85% of the input, respectively. Molecular weights are in kD.

[1995](). Binding of the W861A mutant of TKS5 to FGD1 was comparable to wild-type TKS5. In contrast, the SH3[#5] (W1092A) mutant retained only 10% of the wild-type TKS5 binding capacity ([Fig. S4]). MDA-MB-231 cells were silenced for endogenous TKS5, resulting in reduced invadopodia formation and activity ([Fig. S1, C–E]), and cells were transfected with TKS5 wild-type or SH3 domain mutant constructs to assess their rescue potential. Wild-type and W861A TKS5 proteins localized to classic elongated invadopodia structures, in which they colocalized with cellular FGD1 ([Fig. 4, D–G]). In contrast, the W1092A substitution in the SH3[#5] domain (as well as the double

W861A/W1092A mutation) severely affected the colocalization of TKS5 with FGD1 ([Fig. 4, D–F]) and strongly reduced the formation of FGD1-positive invadopodia in association with the matrix fibers ([Fig. 4 G]). Altogether, these data demonstrated that the integrity of the SH3[#5] domain of TKS5 and its capacity to interact with FGD1 were required for FGD1 recruitment and invadopodia formation. In reciprocal experiments, we observed that overexpression of FGD1[GFP] could partially rescue the absence of endogenous FGD1 protein in silenced cells based on the quantification of TKS5-positive elongated structures ([Fig. 4, H and I]). In contrast, a truncated FGD1 protein with a deletion of the amino-terminal PRD region did not rescue TKS5 recruitment ([Fig. 4, H and I]). Collectively, these data demonstrated that the interaction of TKS5 with FGD1 is required for invadopodia formation and function.

## Identification of a polarity TKS5/FGD1/CDC42 axis required for the 3D collagen invasion program of breast tumor cells

FGD1 is a GEF specific for the Rho GTPase, CDC42. Silencing of CDC42 with two independent siRNAs strongly impaired the formation of invadopodia and collagen degradation by MDA-MB-231 cells ([Fig. 5, A and B]; [Fig. S2 D]; and [Fig. S3 D]), confirming that CDC42 is a master regulator of invadopodia formation and activity in cancer cells ([Ayala et al., 2009]; [Di Martino et al., 2014]; [Sakurai-Yageta et al., 2008]; [Yamaguchi et al., 2005]).

We recently reported that MT1-MMP localizes in late endosomal/lysosomal (LE/Lys) compartments, which congregate near the centrosome located ahead of the nucleus during the invasion of MDA-MB-231 cells in a dense 3D collagen fibrillary matrix ([Infante et al., 2018]). In addition, we found that the defective polarization of MT1-MMP-positive LE/Lys impairs the degradative and invasive potential of breast cancer cells in 3D collagen, probably by interfering with MT1-MMP recycling from storage LE/Lys compartments to the invadopodial plasma membrane ([Infante et al., 2018]). Live-cell real-time confocal spinning disk microscopy of MDA-MB-231 cells expressing mCherry-tagged MT1-MMP and H2B[GFP] in a 3D collagen gel confirmed the polarization of MT1-MMP-positive LE/Lys in front of the nucleus in the direction of movement ([Fig. 5 C]). Next, we investigated the consequences of TKS5, FGD1, or CDC42 silencing on MT1-MMP-containing LE/Lys distribution by automatic tracking over the time as previously described ([Infante et al., 2018]). We confirmed that MT1-MMP[mCh]-containing LE/Lys were highly polarized in front of the nucleus in MDA-MB-231 cells invading through a dense collagen gel ([Fig. 5, C–F]). In addition to the strong impairment of invadopodia formation and collagenolysis along with decreased invasion capacity described above, silencing of TKS5 completely disrupted the polarity of MT1-MMP-positive LE/Lys, which were randomly distributed relative to the direction of nuclear movement ([Fig. 5, E and F]). In addition, the speed of nuclear (cell) movement was strongly reduced in TKS5-depleted cells ([Fig. 5 G]), in agreement with the observed requirement for TKS5 during 3D invasion ([Fig. 1, E and F]). Thus, we concluded that TKS5 was required for LE/Lys and cell polarity that was indispensable for 3D invasion. Similarly, silencing of FGD1 led to a strong defect of polarization of MT1-MMP-containing LE/Lys ([Fig. 5, H and I]). Noticeably, tracking

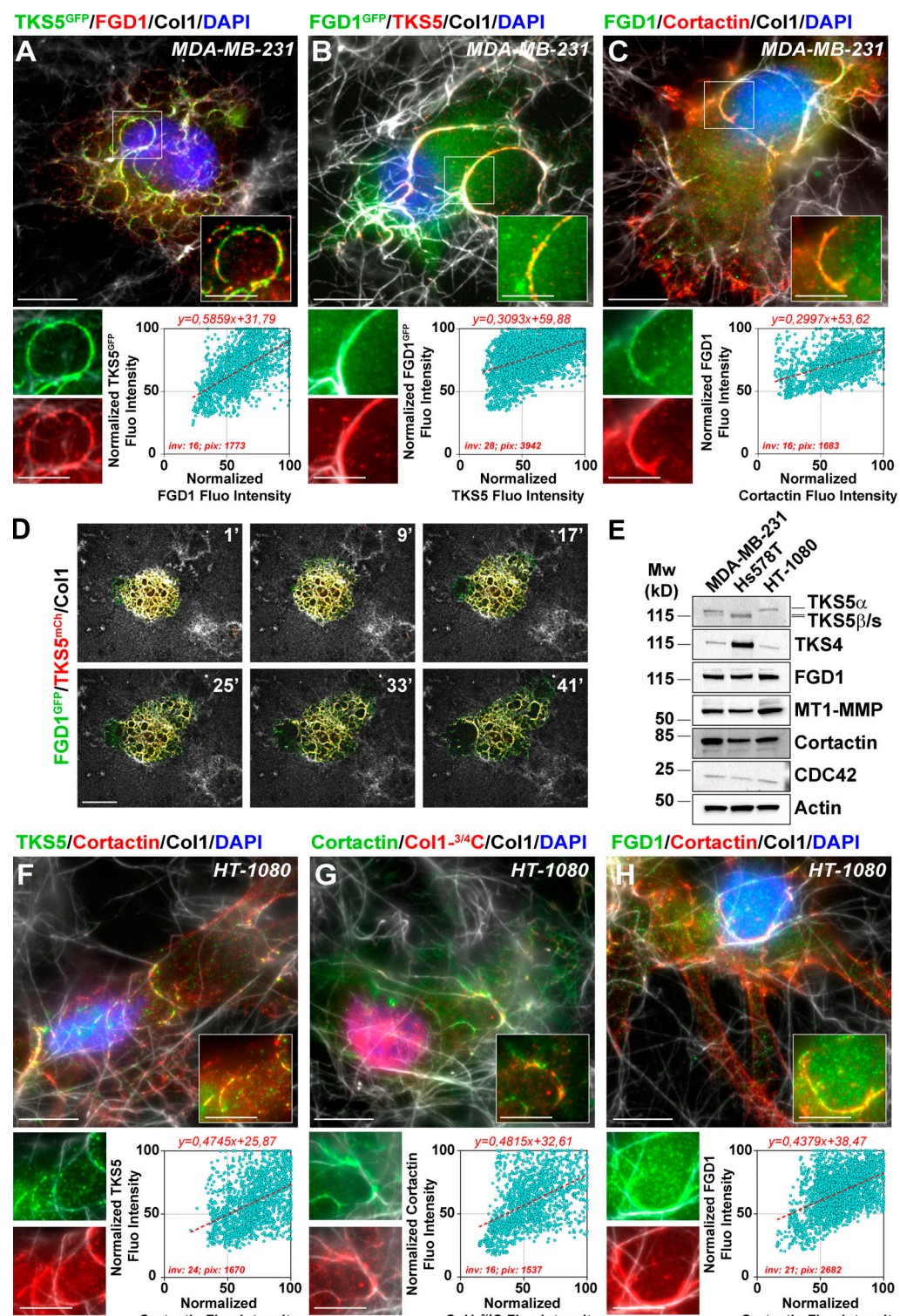

Figure 3. **FGD1 is enriched in collagenolytic invadopodia. (A)** Upper panel: Cells transfected with TKS5[GFP] (green) were seeded on type I collagen (gray) for 60 min and stained with FGD1 antibody (red) and DAPI (blue). Scale bar, 10 µm; zoom-in of boxed region, 5 µm. Lower panel: Correlation of FGD1 (x axis) and TKS5[GFP] (y axis) pixel normalized fluorescence (Fluo) intensity along 16 invadopodial structures. inv, number of invadopodia analyzed; pix, total number of pixels analyzed for both markers. **(B)** Cells transfected with FGD1[GFP] (green) were seeded on type I collagen (gray) and stained with TKS5 antibody (red), and nuclei were stained with DAPI (blue). Scale bar, 10 µm; zoom-in of boxed region, 5 µm. The lower panel shows the correlation of TKS5 and FGD1[GFP] pixel fluorescence intensity as in A. **(C)** Cells seeded on type I collagen (gray) were stained for FGD1 (green) and cortactin (red) and with DAPI (blue). Scale bar, 10 µm; zoom-in of boxed region, 5 µm. The lower panel shows the correlation of cortactin and FGD1 pixel fluorescence intensity as in A. **(D)** The gallery shows nonconsecutive frames from a representative video of a MDA-MB-231 cell expressing FGD1[GFP] and TKS5[mCherry] on a type I collagen fibrillar layer. The time-lapse sequence was started 45 min after plating the cells on type I collagen (see Video 1). Scale bar, 10 µm. **(E)** Immunoblotting analysis of MDA-MB-231,

Hs578T and HT-1080 cell lysates with the indicated antibodies. Loading was controlled using actin antibody. **(F–H)** HT-1080 fibrosarcoma cells were plated on type I collagen (gray) and stained for the indicated invadopodia components. Scale bars, 10 µm; zoom-in of boxed regions, scale bars, 5 µm. The lower panels show the correlation of normalized fluorescence intensity for the indicated markers along the invadopodial structures as in A.

of FGD1-depleted cells revealed that although nuclear (cell) speed was not significantly reduced compared with control cells, cells were unable to maintain their direction of motion (reduced cell persistence), indicative of a lack of polarity and effective cell movement (Fig. 5, J and K). Finally, we observed that silencing of CDC42 also disrupted the polarity of MT1-MMP–positive LE/Lys (Fig. 5, L and M) and globally reduced cell speed, although some of these effects were dependent on the siRNA used (Fig. 5 N). Altogether, our data implicated a TKS5/FGD1/CDC42 axis in the polarization of MT1-MMP storage vesicles required for the persistent and efficient migration of breast cancer cells in a dense 3D matrix environment.

### TKS4 invadopodia in Hs578T breast cancer cells function independently of the FGD1/CDC42 cascade

As mentioned above, immunoblotting analysis revealed distinct TKS5 expression profiles in invasive MDA-MB-231 and HT-1080 versus Hs578T cancer cells (Fig. 3 E; Hughes et al., 2008). Whereas MDA-MB-231 and HT-1080 cells predominantly expressed a high molecular weight TKS5 isoform (designated TKS5α), Hs578T cells mainly expressed two lower molecular weight species previously designated TKS5β and TKS5s ("short") isoforms, which do not contain the amino-terminal PX domain and have lost functionality associated with membrane localization (Fig. 3 E; Saini and Courtneidge, 2018). In addition, MDA-MB-231 and HT-1080 cells expressed low levels of the related TKS4 protein. In contrast, high TKS4 levels were expressed by Hs578T cells, whereas expression of other key invadopodial proteins, including MT1-MMP, cortactin, FGD1, and CDC42, were similar in all three cell lines (Fig. 3 E). Interestingly, it has been reported that only the PX domain–containing TKS5α can contribute to metastasis and invadopodia formation in lung adenocarcinoma (Li et al., 2013), raising the questions whether and how Hs578T cells can form functional invadopodia in the absence of the highest molecular weight TKS5α isoform, and whether TKS4 could substitute. To answer these questions, we first investigated whether TKS4, similar to TKS5, could interact with FGD1. All endogenously expressed TKS5 isoforms in the two cell lines were coimmunoprecipitated with overexpressed FGD1[GFP], whereas TKS4 was not (Fig. 6 A). Similarly, FGD1 was coimmunoprecipitated with TKS5[GFP] but not with TKS4[GFP] (Fig. S5 A). Thus, these data demonstrated that Hs578T cells express high levels of the TKS4 protein, which does not interact with FGD1.

Immunofluorescence labeling of MDA-MB-231 cells revealed that TKS4 was associated with cortactin-positive invadopodia forming in association with collagen fibrils (Fig. 6 B), similar to TKS5. However, knockdown of TKS4 did not interfere with the formation of (TKS5-positive) invadopodia in these cells (Fig. 6 C and Fig. S2 E). In Hs578T cells, expressing high levels of TKS4, this protein localized to cortactin-positive structures that

formed in association with the collagen fibrils (Fig. 6 D). However, cortactin- and TKS4-positive structures were usually shorter and straighter compared with typical curvilinear cortactin- and TKS5-positive invadopodia in MDA-MB-231 cells. Quantification of the area covered by TKS4- or TKS5-positive structures over the whole cell surface revealed that TKS4-positive invadopodia in Hs578T cells represented only ~40% of the area of TKS5-positive invadopodia in MDA-MB-231 cells (not depicted). In addition, TKS5β/s isoforms were barely detectable in cortactin-positive structures in Hs578T cells (Fig. 6 E), suggesting that PX domain–truncated isoforms of TKS5 do not contribute to invadopodia formation in this cell line. Moreover, we observed that silencing of CDC42 significantly affected neither collagen degradation nor TKS4 accumulation in linear structures forming in association with collagen fibrils (Fig. 6, F and G; and Fig. S2 F). Of note, we could not analyze the effect of TKS4 knockdown in Hs578T cells owing to high expression level and resistance to TKS4 mRNA silencing. Therefore, we concluded that although all cell lines used in this study could degrade type I collagen (Fig. S5 B), they did so by deploying different mechanisms: MDA-MB-231 and HT-1080 cells relied on a TKS5/FGD1/CDC42 axis, whereas Hs578T cells used a TKS4-dependent, TKS5-, CDC42-, and FGD1-independent mechanism to assemble collagenolytic invadopodia.

## Discussion

In the podosome/invadopodia protease-dependent program of tissue invasion, actin polymerization is instrumental in generating protrusive forces as a means of maintaining a tight apposition of the MT1-MMP–based pericellular proteolytic machinery with the surrounding ECM fibers and of widening preexisting matrix pores (Dalaka et al., 2020; Ferrari et al., 2019a, 2019b; Infante et al., 2018; Wolf et al., 2013). In addition, direct binding of MT1-MMP to invadopodial F-actin through its 20-aa-long cytosolic tail is thought to anchor MT1-MMP to the invadopodial plasma membrane domain (Yu et al., 2012). The assembly of the invadopodial actin core structure requires the Rho GTPase CDC42, which controls the actin nucleating Arp2/3 complex by activating the Arp2/3 activator, N-WASP (Desmarais et al., 2009; Eddy et al., 2017; Juin et al., 2012, 2014; Monteiro et al., 2013; Yamaguchi et al., 2005; Yu et al., 2012). Similarly, the CDC42/N-WASP/Arp2/3 branched actin assembly module is implicated in the formation of invadopodia-related podosomes, which are required for tissue infiltration by cells of the innate immune system and in endothelial cells during blood vessel remodeling (Linder et al., 2011; Varon et al., 2006). Additionally, the related scaffold TKS5 and TKS4 proteins are indispensable for invadopodia assembly and function (Buschman et al., 2009; Seals et al., 2005). Earlier work has established that TKS5 can interact with cortactin and N-WASP, linking TKS5 to regulation of the branched actin

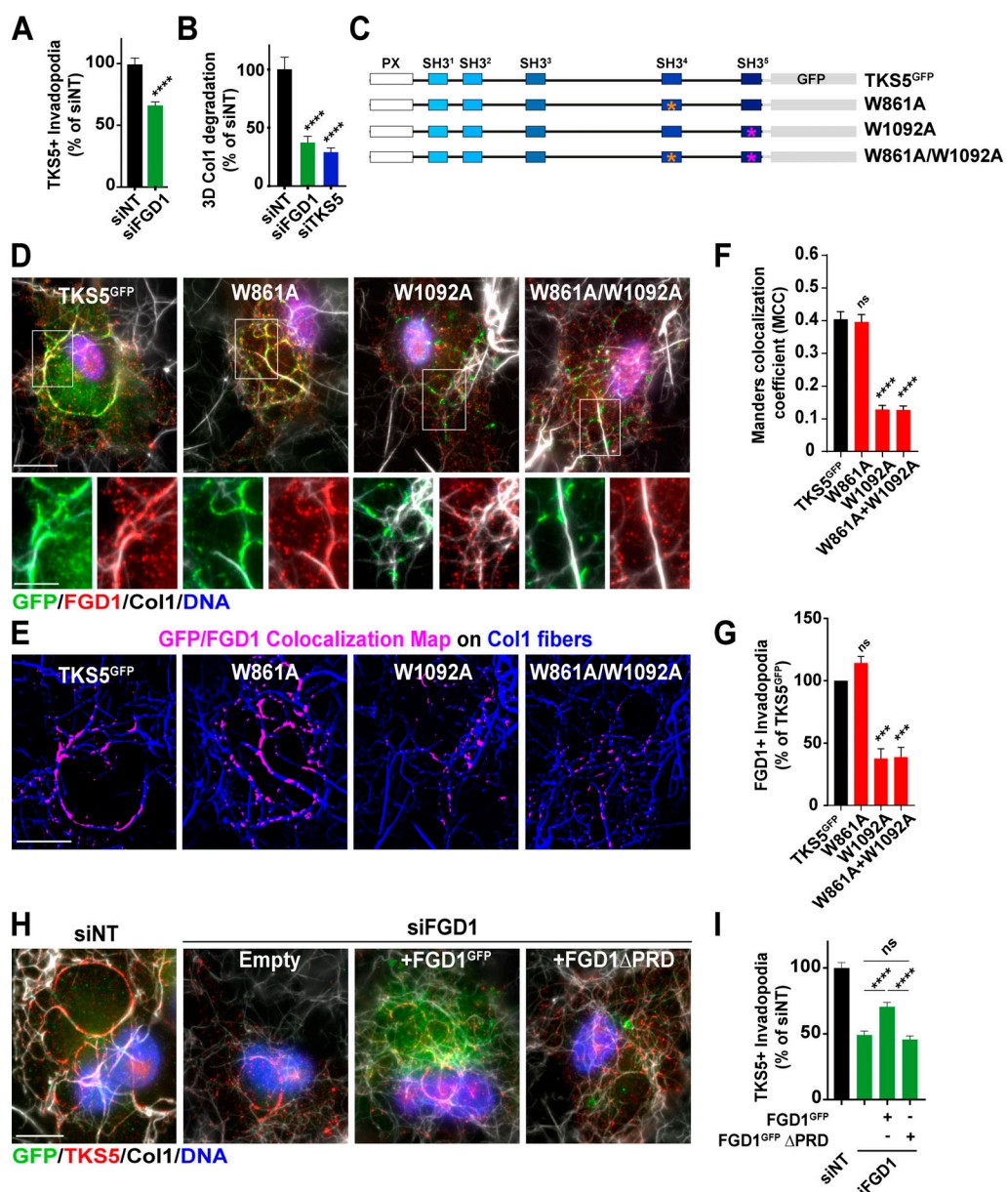

Figure 4. **TKS5 interaction with FGD1 is required for invadopodia formation. (A)** Quantification of TKS5-positive invadopodia in siRNA-treated MDA-MB-231 cells. The y axis is the ratio of the TKS5 area to the total cell area normalized to the mean value of control siNT-treated cells (as percentage ± SEM). siNT, $n$ = 127 cells; siFGD1, $n$ = 146 cells, from three independent experiments. Mann-Whitney test. **(B)** Pericellular collagenolysis by MDA-MB-231 cells treated with the indicated siRNAs measured as mean intensity of Col1-$^{3/4}$C signal per cell. Values for control cells were set to 100%. siNT, $n$ = 34 cells; siFGD1, $n$ = 30 cells; siTKS5, $n$ = 28 cells, from three independent experiments. Kruskal-Wallis test. **(C)** Position of W-to-A substitutions in TKS5 SH3$^{#4}$ (W861A) and SH3$^{#5}$ (W1092A) domains. **(D)** MDA-MB-231 cells were knocked down for endogenous TKS5 upon siRNA treatment, and transfected with the indicated TKS5$^{GFP}$ constructs. Cells were plated on type I collagen (gray) and stained for GFP tag (green) and FGD1 (red) and nuclei were stained with DAPI (blue). Scale bar, 10 μm; zoom-in of boxed region, scale bar, 5 μm. **(E)** TKS5$^{GFP}$/FGD1 colocalization map (magenta) in the different cell populations as in D. The underlying collagen network is shown in blue. Scale bar, 10 μm. **(F)** TKS5$^{GFP}$/FGD1 Manders correlation coefficient in the different cell populations as in D. Tukey's multiple comparisons test. **(G)** Quantification of FGD1-positive invadopodia in cells silenced for endogenous TKS5 as in D and transfected with the indicated TKS5$^{GFP}$ constructs as in D. The y axis is the ratio of the FGD1 area to the total cell area normalized to the mean value of control siTKS5/TKS5$^{GFP}$-treated cells (as percentage ± SEM). TKS5$^{GFP}$, $n$ = 47 cells; W861A, $n$ = 50 cells; W1092A, $n$ = 53 cells; W861A+W1092A, $n$ = 43 cells from three independent experiments. Tukey's multiple comparisons test. **(H)** MDA-MB-231 cells were knocked down for endogenous FGD1 upon siRNA treatment, and transfected with the indicated FGD1$^{GFP}$ constructs. Cells were plated on type I collagen (gray) and stained for Tks5 (red) and nuclei were stained with DAPI (blue). Scale bar, 10 μm. **(I)** Quantification of TKS5-positive invadopodia in the different cell populations. The y axis is the ratio of the TKS5 area to the total cell area normalized to the mean value of control siNT-treated cells (as percentage ± SEM). siNT, $n$ = 75 cells; siFGD1, $n$ = 74 cells; siFGD1/FGD1$^{GFP}$, $n$ = 98 cells and siFGD1/FGD1$^{GFP}$ΔPRD, $n$ = 67 cells from three independent experiments. Tukey's multiple comparisons test. ***, P < 0.001; ****, P < 0.0001; ns, not significant.

Figure 5. **A TKS5/FGD1/CDC42 axis is required for cell polarization during 3D collagen invasion. (A)** Quantification of TKS5-positive invadopodia in siRNA-treated MDA-MB-231 cells as indicated. siNT, $n = 238$ cells; siCDC42#04, $n = 88$ cells; siCDC42#07, $n = 48$ cells and siMT1-MMP, $n = 68$ cells from three independent experiments. **(B)** Pericellular collagenolysis by MDA-MB-231 cells treated with the indicated siRNAs. siNT, $n = 50$ cells; siCDC42#04, $n = 21$ cells; siCDC42#07, $n = 25$ cells and siMT1-MMP, $n = 16$ cells from three independent experiments. Statistical analysis in A and B was based on Kruskal-Wallis test. **(C)** MDA-MB-231 cells expressing MT1-MMP$^{mCh}$ (red) and H2B$^{GFP}$ (green) were embedded in the 3D collagen gel (blue) and analyzed by real-time spinning disk confocal microscopy. The gallery shows nonconsecutive frames from a representative video obtained from three independent experiments (time in hour: minute). Arrows show the direction of movement from the previous image. The cyan dotted lines represent the cell track over time. The position of the nucleus in the previous image is shown by a yellow dotted line with an arrow representing the nucleus movement. Scale bar, 10 μm. **(D)** MDA-MB-231 cell expressing MT1-MMP$^{mCh}$ (red) and H2B$^{GFP}$ (green) embedded in the 3D collagen gel (blue). The cell is divided in 30° sectors, with the 0° axis representing the direction of nucleus movement. **(E, H, and L)** Rose plots showing the percentage of MT1-MMP$^{mCh}$-positive LE/Lys in 30° sectors relative to the direction of nucleus movement (0°) scored from time-lapse sequences of MT1-MMP$^{mCh}$/H2B$^{GFP}$-expressing MDA-MB-231 cells or cells treated with TKS5 or FGD1 siRNA or control siNT siRNA invading in the 3D collagen gel. c, number of cells; e, number of endosomes analyzed from three independent experiments. P values for circular uniformity Rao's spacing tests are shown. **(F, I, and M)** Percentage of MT1-MMP$^{mCh}$-positive LE/Lys in a 120° sector in front of the nucleus in the direction of movement (as in D). Data were analyzed using $t$ test (F and I) or one-way ANOVA (M). **(G, J, and N)** H2B$^{GFP}$-positive nuclei were automatically tracked from the time-lapse sequences obtained from three independent experiments, and the plots show the distribution of nuclei speed. Data were analyzed using $t$ test (G and J) or one-way ANOVA (N). **(K)** Persistence of nucleus movement computed from the H2B$^{GFP}$-positive nuclei trajectories. Data were analyzed using $t$ test. ***, $P < 0.001$; ****, $P < 0.0001$; ns, not significant.

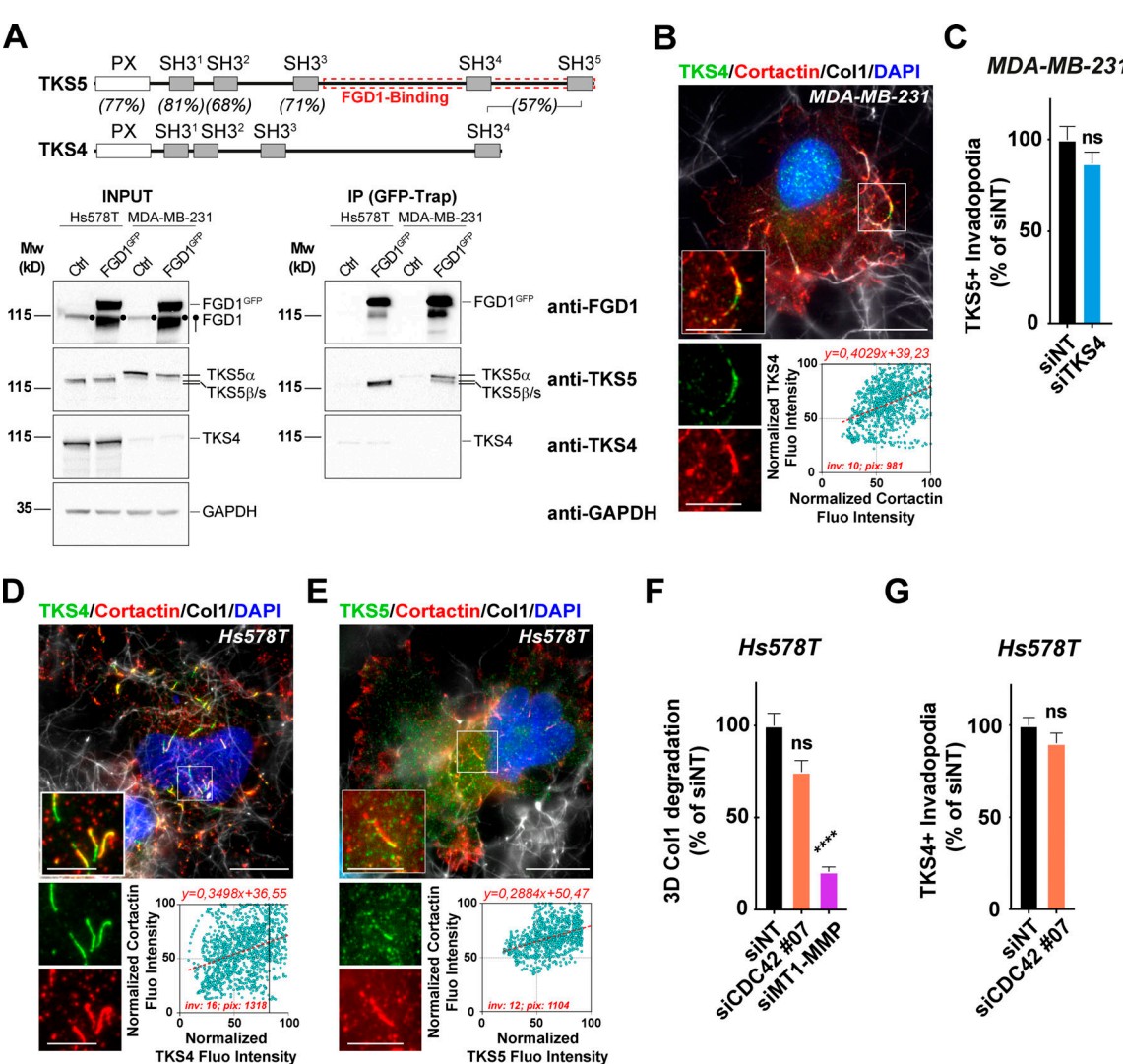

Figure 6. **The mechanism of invadopodia formation directed by TKS4 is independent of FGD1 and CDC42. (A)** Schematic representation of TKS5 and TKS4 domains and percentage of identity in the different domains (Saini and Courtneidge, 2018). Lysates of Hs578T or MDA-MB-231 human breast cancer cells expressing FGD1^GFP or not (Ctrl) were immunoprecipitated with GFP antibodies. Bound proteins were analyzed with FGD1, TKS4, and TKS5 antibodies (IP). 5% of total lysate was loaded as a control (input). Equal loading was controlled using GAPDH antibody. Molecular weights are in kD. **(B)** MDA-MB-21 cells seeded on type I collagen (gray) were stained for TKS4 (green) and cortactin (red), and nuclei were stained with DAPI (blue). Scale bar, 10 µm; zoom-in of boxed region, 5 µm. Lower panel: Correlation of cortactin (x axis) and TKS4 (y axis) pixel normalized fluorescence intensity along the invadopodial structures. inv, number of invadopodia analyzed; pix, total number of pixels analyzed for both markers. **(C)** Quantification of TKS5-positive invadopodia in MDA-MB-231 cells silenced for TKS4. The y axis is the ratio of the TKS5 area to the total cell area normalized to the mean value of control siNT-treated cells (as percentage ± SEM). Mann–Whitney U test. **(D and E)** Hs578T cells seeded on type I collagen (gray) were stained for TKS4 (green, D) or TKS5 (green, E), cortactin (red), and DAPI (blue). Boxed regions highlight cortactin-positive invadopodia. Scale bar, 10 µm; zoom-in of boxed regions, 5 µm. The lower panels show the correlation of pixel normalized fluorescence intensity for the indicated markers along the invadopodial structures as in B. inv, number of invadopodia analyzed, pix, total number of pixels analyzed for both markers. **(F)** Quantification of pericellular collagenolysis by Hs578T cells treated with CDC42 or MT1-MMP siRNAs measured as mean intensity of Col1-3/4C signal per cell. Values for control cells (siNT) were set to 100%. siNT, n = 53 cells; siCDC42#07, n = 40 cells; and siMT1-MMP, n = 53 cells from three independent experiments. Kruskal–Wallis test. **(G)** Quantification of TKS4-positive invadopodia in Hs578T cells silenced for CDC42. siNT, n = 266 cells; siCDC42#07, n = 125 cells from three independent experiments. Statistical analysis is based on a Kruskal–Wallis test. ****, P < 0.0001; ns, not significant.

meshwork (Lynch et al., 2003; Stylli et al., 2009). However, little is known about signaling pathways that lead to invadopodia assembly downstream of TKS proteins. In addition, interrelation between the TKS and CDC42 signaling modules is not known.

Here, we show that TKS5 is required for the assembly and matrix-degradative function of elongated F-actin/cortactin-positive invadopodia forming in association with type I collagen fibers. Our coimmunoprecipitation and MS experiments

identify an interaction between TKS5 and the CDC42-GEF, FGD1 (Pasteris et al., 1994; Zheng et al., 1996). The TKS5/FGD1 interaction, which was initially reported in a global human interactome study (Hein et al., 2015), is now validated by the present work in the context of stromal remodeling by tumor cells. We show that the carboxy-terminal region of TKS5 (aa 505–1118), encompassing two of five SH3 domains (SH3#4 and SH3#5), interacts with the amino-terminal proline-rich domain

of FGD1 (aa 1–335). Specifically, we found that mutation of a critical W residue in SH3[#5] (Tanaka et al., 1995) abolishes binding of TKS5 to FGD1 and inhibits FGD1 recruitment and invadopodia formation in cells deleted for the endogenous TKS5 protein. In contrast, the related TKS4 protein, although it also localizes to invadopodia, is unable to interact with FGD1. Of note, while the identity between the conserved PX and SH3 amino-terminal domains of TKS5 and TKS4 is relatively high (∼68–84%; Fig. 6 A), the overall identity drops to ∼55% between the FGD1-binding region of TKS5 (aa 505–1118) and the corresponding carboxy-terminal region of TKS4, probably accounting for the lack of interaction between TKS4 and FGD1. Therefore, in MDA-MB-231 (and HT-1080) cells, TKS5α drives invadopodia formation in association with FGD1/CDC42, whereas in Hs578T cells that express shorter, PX domain-truncated, isoforms of TKS5 (TKS5β/s), TKS4 directs the formation of collagenolytic invadopodia in a CDC42- and FGD1-independent manner. We noticed that Hs578T breast cancer cells formed relatively straight TKS4-positive invadopodia, in contrast to the typical curvilinear TKS5-positive invadopodia in MDA-MB-231 cells (compare Figs. 1 A and 6 D). Whether these differences in TKS5/FGD1/CDC42 dependence and invadopodia morphology underlie functional and/or biomechanical disparities of TKS4- versus TKS5-positive invadopodia is an intriguing possibility. Along this line, we recently proposed a critical contribution of the curvature of TKS5-positive invadopodia to the mechanism of force production. Our model is that frictional forces can appear in the invadopodial actin meshwork owing to the curved geometry of the invadopodia/collagen fiber ensemble, thus allowing growing actin filaments to push back on neighboring filaments and push forward the collagen fiber for matrix pore widening (Ferrari et al., 2019b). This model may not apply in the case of TKS4-positive, TKS5-negative cells such as Hs578T cells, which form straight invadopodia with a reduced pushing capacity and are endowed merely with collagenolytic function.

Loss-of-function mutations in the gene encoding FGD1 cause faciogenital dysplasia, a rare X-linked disorder that manifests in defects of bone development, craniofacial abnormalities, and mental retardation (Pedigo et al., 2016). FGD1 is also essential for CDC42-dependent podosome and invadopodia assembly (Ayala et al., 2009; Daubon et al., 2011). Our data demonstrate that the essential function of TKS5 in directing the formation of collagenolytic invadopodia in breast cancer cells requires FGD1, probably through the direct modulation of CDC42 activity. FGD1, which interacts with cortactin, may also regulate Arp2/3 complex–mediated actin assembly independently of its catalytic activity (Kim et al., 2004). Remarkably, an interaction between activated CDC42 and TKS5 was recently reported based on a proximity-dependent biotinylation approach, identifying TKS5 as a novel bona fide CDC42 effector protein (Bagci et al., 2020). In addition, we identified the CDC42 effector proteins, formin-binding protein 17 (FBP17) and formin-binding protein 1-like (TOCA1), as TKS5 partners in our coimmunoprecipitation experiments (Table S1). Both FBP17 and TOCA1 interact with and activate the N-WASP/WASP-interacting protein complex, which is competent for Arp2/3 activation and thus may potentiate the

induction of invadopodial branched actin assembly downstream of TKS5 (Suetsugu and Gautreau, 2012). Also of note is the recent finding that FBP17 interacts with and recruits the 5′-inositol phosphatases SHIP1 and SHIP2, the main enzymes involved in $PI(3,4)P_2$ production from plasma membrane phosphatidylinositol (3,4,5)-trisphosphate (Chan Wah Hak et al., 2018; Xiong et al., 2016). In addition, SHIP2 binds to the mechanosignaling protein p130[CAS] (Prasad et al., 2001), which can interact with the cytosolic tail of MT1-MMP (Gingras et al., 2008; Gonzalo et al., 2010). We observed that SHIP2 and p130[CAS] accumulated together with cortactin in invadopodia in association with the collagen fibers (Fig. 7, A and B). In addition, overexpressed FLAG-tagged inositol polyphosphate 4-phosphatase type II, an enzyme that catabolizes $PI(3,4)P_2$ into $PI(3)P$, was enriched in TKS5-positive invadopodia, probably by binding to its $PI(3,4)P_2$ substrate (Fig. 7 C). Thus, a network of interactions is emerging that potentially link surface-exposed MT1-MMP, at contact sites with collagen fibers in the matrix, to an early and local production of (SHIP2-mediated) $PI(3,4)P_2$, which may recruit TKS5 (through its PX domain) and insert more MT1-MMP in the invadopodial plasma membrane in a feed-forward invadopodia maturation loop (Fig. 7 D). Along this line, SHIP2 was shown to promote invadopodia formation and maturation (Eddy et al., 2017; Hoshino et al., 2012; Sharma et al., 2013). Collectively, these data suggest that a MT1-MMP/p130[CAS]/SHIP2 interaction network may position MT1-MMP as an initiator of the invadopodial response by ensuring $PI(3,4)P_2$ production at early contact sites between surface-exposed MT1-MMP and type I collagen fibrils, preceding the expansion and stabilization phase of invadopodia structures (Fig. 7 D). The recently uncovered role of MT1-MMP in directing invadopodia assembly independent of its proteolytic activity goes along this line (Ferrari et al., 2019b).

Our study also unveils a novel cell polarity module comprising the TKS5/FGD1 complex and activated CDC42. Loss of function of any of these components drastically prevents the polarization of recycling MT1-MMP–containing vesicles ahead of the nucleus in cancer cells invading through the confining matrix meshwork. Consequently, assembly of collagenolytic invadopodia and matrix degradation are prevented, and the directionality and speed of tumor cell invasion through the ECM are reduced. Interestingly, TKS5 shares a conserved domain organization with the PX and SH3 domain–containing protein, Bem1p, a scaffold protein essential for the establishment of cell polarity in *Saccharomyces cerevisiae*. Similar to TKS5, which interacts with active CDC42 (Bagci et al., 2020) and with its GEF, FGD1 (this study), Bem1p interacts with GTP-bound Cdc42p, and it binds the CDC42 GEF, Cdc24p, thus serving as platform for local self-amplification of CDC42 activity (Martin, 2015). Disruption of the Bem1 interaction module abolishes polar bud growth in *S. cerevisiae* (Martin, 2015), similar to the loss of TKS5 that prevents the formation of collagenolytic invadopodia. Thus, an intriguing possibility to be tested in the future is that tumor cells may use an ancient conserved polarity module for the amplification of small initial stimuli resulting from cell/ECM recognition and mechanosignaling to promote the formation of self-assembling mature matrix-degrading invadopodia through positive feedback mechanisms (Fig. 7 D; McCusker, 2020).

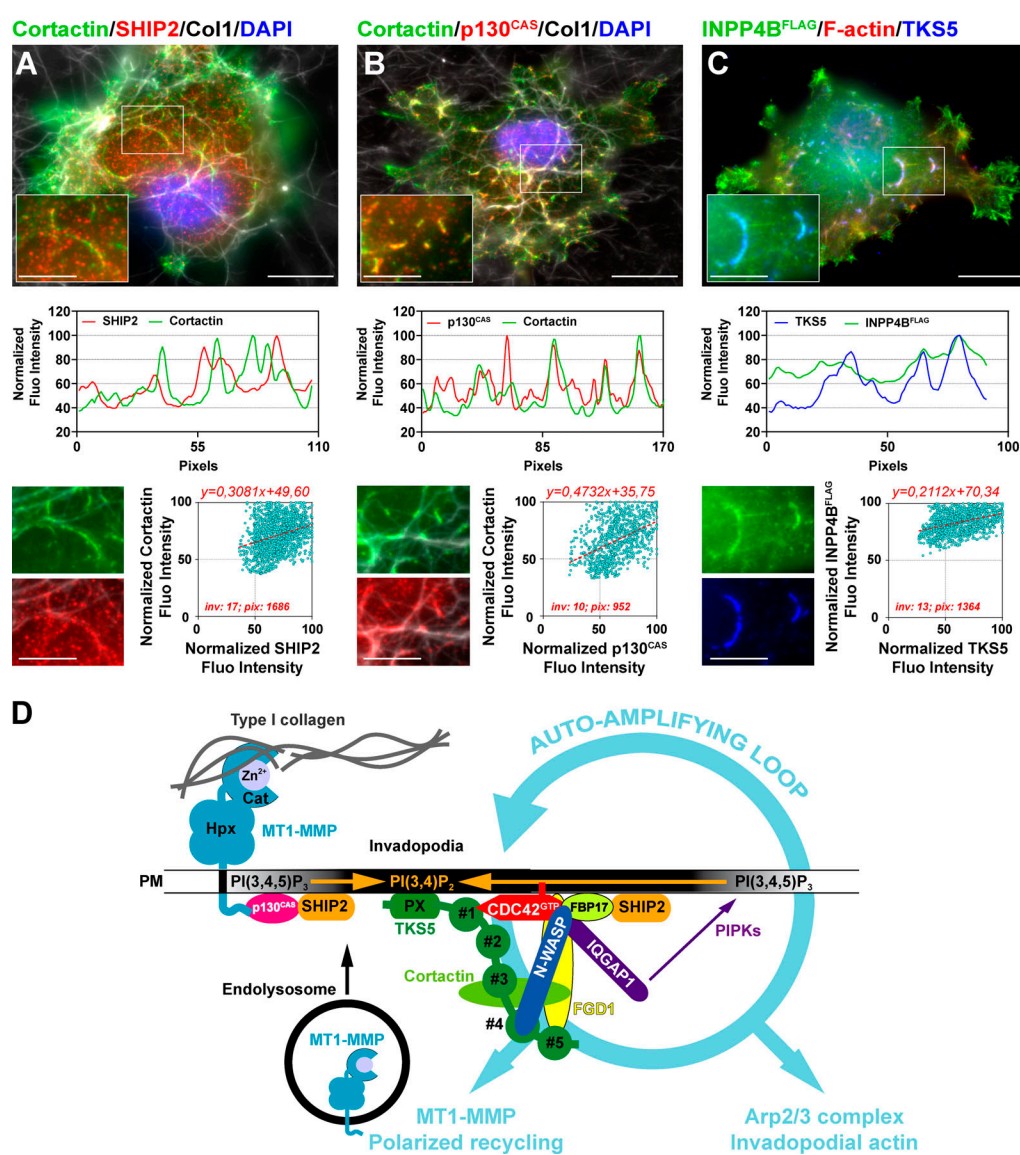

Figure 7. **A TKS5/FGD1/CDC42 signaling module directs the assembly of collagenolytic invadopodia in breast cancer cells. (A)** Upper panel: Cells were plated on a layer of fibrillary collagen (gray) and stained for SHIP2 (red) and cortactin (green), and nuclei were stained with DAPI (blue). Scale bar, 10 µm; zoom-in of boxed region, 5 µm. Middle panel: Fluorescence (Fluo) intensity profiles for SHIP2 and cortactin were recorded along an invadopodium (boxed region in upper panel) and normalized to the maximum fluorescence intensity set to 100. Lower panel: Correlation of SHIP2 (x axis) and cortactin (y axis) fluorescence intensity along the invadopodial structures. inv, number of invadopodia analyzed; pix, total number of pixels analyzed for both markers. **(B)** Correlation of p130[CAS] (red) and cortactin (green) as in A. **(C)** Correlation of TKS5 (blue) and overexpressed FLAG-tagged inositol polyphosphate 4-phosphatase type II (green) as in A. **(D)** Model of feed-forward loop self-assembly mechanisms for invadopodia formation and maturation. Surface-exposed MT1-MMP in association with the collagen fibers within the matrix may contribute to the early production of PI(3,4)P$_2$, the phosphoinositide ligand of the TKS5 PX domain, by interacting with a p130[CAS]/SHIP2 complex. Accumulation of PI(3,4)P$_2$ at collagen–cell contact sites leads to the recruitment of TKS5 to the forming invadopodial plasma membrane. The carboxy-terminal SH3 domains of TKS5 interact with and recruit the highly specific CDC42-GEF, FGD1. GTP-bound active CDC42 interacts with several downstream effectors, which are known to contribute to invadopodia assembly and function including TKS5 itself, N-WASP, the polarity protein, IQGAP1, and FBP17. Arp2/3 complex activation by N-WASP leads to the assembly of the invadopodia branched actin core structure. FBP17 also positively influences branched actin assembly and may lead to PI(3,4)P$_2$ production by interacting with SHIP2. Recruitment of IQGAP1 at invadopodia contributes to the polarized recycling of MT1-MMP from endolysosomal compartments (Sakurai-Yageta et al., 2008). In addition, IQGAP1 scaffolds a network of phosphoinositide kinases, potentially leading to further phosphatidylinositol (3,4,5)-trisphosphate and PI(3,4)P$_2$ accumulation (Choi et al., 2016). This network of interactions further promotes invadopodia assembly and function in a feed-forward loop with contributions of actin assembly and membrane trafficking. Cat, catalytic domain; Hpx, hemopexin domain; PIPK, phosphoinositide kinase; PM, plasma membrane. The PX and SH3 domains #1 to #5 of TKS5 are depicted.

Altogether, our work highlights a new, potentially druggable, molecular polarity feed-forward loop framework based on the interplay of TKS5, FGD1, and CDC42 function in the regulation of invadopodia activity as an essential axis of the MT1-MMP–based metastatic program of breast cancer cells.

## Materials and methods
### Antibodies and reagents

The source and working dilution of commercial antibodies used for this study are listed in Table S2. Hepatocyte growth factor (HGF) was purchased from PeproTech and used at 20 ng/ml.

## DNA constructs

Plasmids used for this study and sources are listed in Table S3. For the cloning of TKS5 truncated constructs, fragments were generated by PCR and inserted in the SacI and SalI sites of the pEGFP-N1 vector. The following primers were used: TKS5_1-139[GFP] forward, 5′-CCGGACTCAGATCTCGAGCTCAAGCTTCG-3′, and reverse, 5′-TTTTTGTCGACTGCAGAATTCTCCTCTTGG AACTGCC-3′; TKS5_1-323[GFP] reverse, 5′-TTTTTGTCGACTGCA GAATTCTCACTGGGCCGGC-3′; TKS5_1-505[GFP] reverse, 5′-TTT TTGTCGACTGCAGAATTCTGGGCCGGGTCAG-3′; TKS5_324-1118[GFP] forward, 5′-AAAAAGAGCTCAAGCTTCGAATTATGGAGA TCATTGG-3′, and reverse, 5′-TTTTTGTCGACTGCAGAATTCTGT TCTTTTTCTCAAGG-3′. Point mutations were created in the fourth and fifth SH3 domains of TKS5 by site-directed muta-genesis (QuickChange II, Stratagene) such that the first in a conserved pair of W residues was converted to A to generate TKS5[GFP]W861A, TKS5[GFP]W1092A, and TKS5[GFP]W861A/W1092A (the position of the mutations is based on human TKS5 con-taining the PX domain). TKS5 SH3[#4] forward primer: 5′-GAA GCAGGAGAGAGCGGGGCGTGGTATGTGAGGTTT-3′, and re-verse primer: 5′-AAACCTCACATACCACGCCCCGCTCTCCTG CTTC-3′; TKS5 SH3[#5] forward primer: 5′-GAGGAACCCTAATGG CGCGTGGTACTGCCAGATC-3′, and reverse primer: 5′-GGCAGT ACCACGCGCCATTAGGGTTCCTCTCCAG-3′. The TKS4[GFP] con-struct was obtained by cloning the insert in the SacI and BamH1 sites of the pEGFP-N1 vector. The sequences corresponding to three regions of FGD1 encompassing the entire FGD1 protein were prepared from the pEGFP-C1-FGD1 construct and cloned into pET-MCn-His-GST vector to make GST–FGD1 fusion pro-teins. The N-term domain (PRD, aa 18–335; forward primer, 5′-CGAATTACCATATGTCAGCGGCAAATACCCCCG-3′, and re-verse primer, 5′-CCGCTCGAGAACCTCCTGGGAGCCAG-3′) was cloned into NdeI and XhoI sites, the middle (DH-PH, aa 335–717; forward, 5′-CTTTACTTCCAGGGCCATATGGTTGACAGTGACCTG GAA-3′, and reverse, 5′-TCTAGACTATTAGGATCCCACGTTTGG AGAGTTAGGG-3′) was cloned into NdeI and BamHI sites, and the carboxy-terminal domain (FYVE+PH, aa 717–920; forward, 5′-CTTTACTTCCAGGGCCATATGGTGGATCTTGGGAAGAGG-3′, and reverse, 5′-TCTAGACTATTAGGATCCACGGCCCGCCCT GCCGAGAAC-3′) was cloned into NdeI and BamHI sites of the receiving vector. The GFP-tagged FGD1 construct deleted of the PRD domain (PRD, aa 1–335) was obtained by site-directed mutagenesis of the pEGFP-C1-FGD1 plasmid by PCR using the following primers: forward, 5′-CTCAAGCTTCGAATGACAGTG ACCTGG-3′, and reverse, 5′-GGTCACTGTCATTCGAAGCTTGAG C-3′. PCR products were digested by Dpn1 followed by trans-formation into XL1-Blue cells. Plasmids isolated from the re-sulting colonies were screened for the desired modification by digestion with AflII and XhoI. All constructs were verified by sequencing before use.

## Cell culture, transient transfection, and siRNA treatment

MDA-MB-231 cells (ATCC; HTB-26) were grown in L15 medium supplemented with 15% FCS and 2 mM glutamine at 37°C in 1% $CO_2$. Hs578T cells (kindly provided by Dr. C. Lamaze, Institut Curie, Paris, France) were grown at 37°C under 5% $CO_2$ in DMEM GlutaMAX (Gibco; Life Technologies) supplemented

with 10% FCS (Gibco; Life Technologies), 5 mM pyruvate (Gibco; Life Technologies), and 1% penicillin-streptomycin (Gibco; Life Technologies). HT-1080 fibrosarcoma cells (ATCC; CCL-121) were grown in DMEM GlutaMAX supplemented with 10% FCS. Cells were routinely tested for mycoplasma contamination. MDA-MB-231 cells stably expressing TKS5[GFP] were generated by lentiviral transduction (Ferrari et al., 2019b). For transient ex-pression, cells were transfected using Lipofectamine 3000 (Thermo Fisher Scientific) for 20 h before analysis according to the manufacturer's instructions, with minor modifications. Briefly, cells were trypsinized, and $1.5 \times 10^5$ cells were plated in a 24-well plate. The transfection mixture containing Lipofect-amine 3000 together with P3000 reagent (Thermo Fisher Sci-entific) and 0.5–1 µg of a plasmid was added immediately after cell plating, and the transfection mixture was replaced with fresh growth medium 15 h later. For RNA interference, MDA-MB-231 cells were treated with the indicated siRNAs (50 or 100 nM) using Lullaby (OZ Biosciences) and analyzed after 72 h of treatment. Hs578T cells were treated with indicated siRNAs using Lipofectamine 3000 for 72 h before analysis according to the manufacturer's instructions, with minor modifications. Briefly, the cells were trypsinized, 40,000 cells were plated per well of 24-well plate, and the Lipofectamine 3000 mixture containing 100 nM of siRNA was added immediately. The transfection mixture was replaced with fresh growth medium 15 h later. siRNAs used for this study are listed in Table S3. For rescue experiments, cells were first seeded and treated with the indicated siRNAs (50 nM) using Lullaby. 24 h later, medium was changed, and cells were transfected with the indicated constructs in a transfection mixture containing Lipofectamine 3000 together with P3000 reagent and 0.5–1 µg of plasmid. The transfection mixture was replaced with fresh growth medium 15 h later. Cells were analyzed 72 h after siRNA transfection.

## GST pull-down assay

Cells were transfected with plasmids encoding the indicated constructs using Lipofectamine 3000 (two 100-mm dishes per condition, $3.7 \times 10^7$ cells per dish), and 20 h later, cells were lysed in NP-40 buffer as described for the immunoprecipitation pro-cedure. GST or GST-tagged FGD1 protein fragments (2 µM) were immobilized onto Glutathione Sepharose 4B beads (GE Health-care; 25 µl of beads per condition) in binding buffer (50 mM Tris-HCl, pH 7.5, 150 mM NaCl, 0.5 mM EDTA, 10 mM $MgCl_2$, 10% glycerol, 0.7% NP-40, 0.5% BSA, and cocktails of protease and phosphatase inhibitors) for 1 h at 4°C under constant agi-tation. The beads were washed three times with the washing buffer and incubated with the cell supernatants for 2 h at 4°C under constant agitation. At the end of the procedure, the beads were washed twice with the washing buffer and then twice with the binding buffer. The bound proteins were eluted in Laemmli buffer by heating to 95°C for 10 min and analyzed by SDS-PAGE.

## Immunoisolation of TKS5[GFP]-bound proteins

Cells stably expressing TKS5[GFP] (MDA-MB-231/TKS5[GFP]) or not (MDA-MB-231) were plated on a layer of rat collagen type I for 90 min (four 100-mm dishes for each replicate, $3.7 \times 10^7$ cells per dish), and cells were collected, lysed in NP-40 buffer (50 mM

Tris-HCl, pH 7.5, 150 mM NaCl, 0.5 mM EDTA, 10 mM MgCl$_2$, 10% glycerol, 60 mM β-glucoside, and 1% NP-40) for 30 min at 4°C under constant agitation. The lysate was centrifuged at 15,000 $g$ for 10 min at 4°C, and the supernatant was incubated with 25 µl of equilibrated control magnetic agarose beads (ChromoTek) for 1 h at 4°C under constant agitation. The precleared lysate was then incubated with 25 µl of equilibrated magnetic agarose beads coupled to anti-GFP nanobodies (GFP-Trap; ChromoTek) for another 1 h at 4°C. At the end of the procedure, GFP-Trap beads were washed three times with washing buffer (50 mM Tris-HCl, pH 7.5, 150 mM NaCl, 0,5 mM EDTA, 10 mM MgCl$_2$, 10% glycerol, and 0.7% NP-40), followed by four washes with dilution buffer (50 mM Tris-HCl, pH 7.5, 150 mM NaCl, 0.5 mM EDTA, 10 mM MgCl$_2$, and 10% glycerol) and collected in 200 µl of dilution buffer for overnight storage at 4°C before MS analysis. In immunoprecipitation experiments followed by immunoblotting analysis, after the final wash, the beads were immediately heated at 95°C for 10 min under constant agitation in Laemmli buffer and stored at –20°C before immunoblotting analysis.

### MS analysis
Proteins on magnetic beads were washed twice with 100 µl of 25 mM NH$_4$HCO$_3$, and on-beads digestion was performed with 0.2 µg trypsin/LysC (Promega) for 60 min in 100 µl of 25 mM NH$_4$HCO$_3$. Sample were then loaded onto a homemade C18 StageTips for desalting. Peptides were eluted using 40/60 MeCN/H$_2$O + 0.1% formic acid and vacuum concentrated to dryness.

Online chromatography was performed with an RSLCnano system (Ultimate 3000, Thermo Fisher Scientific) coupled online to an Orbitrap Fusion Tribrid mass spectrometer (Thermo Fisher Scientific). Peptides were trapped on a C18 column (75-µm inner diameter × 2 cm; nanoViper Acclaim PepMapTM 100; Thermo Fisher Scientific) with buffer A (2/98 MeCN/H$_2$O in 0.1% formic acid) at a flow rate of 4.0 µl/min over 4 min. Separation was performed on a 50 cm × 75 µm C18 column (nanoViper Acclaim PepMapTM RSLC, 2 µm, 100 Å, Thermo Fisher Scientific) regulated to a temperature of 55°C with a linear gradient of 5–25% buffer B (100% MeCN in 0.1% formic acid) at a flow rate of 300 nl/min over 100 min. Full-scan MS was acquired in the Orbitrap analyzer with a resolution set to 120,000, and ions from each full scan were fragmented by high-energy collisional dissociation and analyzed in the linear ion trap.

For identification, the data were searched against the human SwisProt database (February 2017) using Sequest$^{HF}$ through Proteome Discoverer (v2.1). Enzyme specificity was set to trypsin, and a maximum of two missed cleavage sites were allowed. Oxidized methionine, amino-terminal acetylation, and carbamidomethyl cysteine were set as variable modifications. Maximum allowed mass deviation was set to 10 ppm for monoisotopic precursor ions and 0.6 D for MS/MS peaks.

The resulting files were further processed using myProMS v3.6 (Poullet et al., 2007). False discovery rate calculation used Percolator and was set to 1% at the peptide level for the whole study. The label-free quantification was performed by peptide extracted ion chromatograms (XICs) computed with MassChroQ

v2.2 (Valot et al., 2011). For protein quantification, XICs from proteotypic peptides shared between compared conditions (TopN matching) with no missed cleavages were used. Median and scale normalization was applied on the total signal to correct the XICs for each biological replicate. To estimate the significance of the change in protein abundance, a linear model (adjusted on peptides and biological replicates) was performed, and P values were adjusted with a Benjamini–Hochberg false discovery rate procedure with a control threshold set to 0.05. The MS proteomics data have been deposited to the ProteomeXchange Consortium via the PRIDE partner repository (Vizcaíno et al., 2016), with the dataset identifier PXD011632.

### Western blot analysis
Cells were lysed in SDS sample buffer, separated by SDS-PAGE, and detected by immunoblotting analysis with the indicated antibodies. Antibodies were visualized using the ECL detection system (GE Healthcare).

### Immunofluorescence analysis of cells plated on fibrillar type I collagen
Coverslips were layered with 200 µl ice-cold 2.0 mg/ml acidic extracted collagen I solution (Corning) in 1× MEM mixed with 4% Alexa Fluor 647–conjugated type I collagen. The collagen solution was adjusted to pH 7.5 using 0.34 N NaOH, and Hepes was added to 25 µM final concentration. After 3 min of polymerization at 37°C, the collagen layer was washed gently in PBS, and cells in suspension were added for 60–90 min at 37°C in 1% CO$_2$ before fixation. Cells were preextracted with 0.1% Triton X-100 in 4% PFA in PBS during 90 s, fixed in 4% PFA in PBS for 20 min, and stained for immunofluorescence microscopy with indicated antibodies.

### Quantification of TKS5- or FGD1-positive invadopodia
$5 × 10^4$ cells were plated on type I collagen–coated coverslips, fixed after 60 min, and stained with TKS5 or FGD1 pAbs and counterstained with cortactin mAb. Images were acquired with a wide-field microscope (Eclipse 90i Upright; Nikon) using a 100× Plan Apo VC 1.4 oil objective and a highly sensitive cooled interlined charge-coupled device camera (CoolSNAP HQ2; Roper Scientific). A z-dimension series of images was taken every 0.2 µm by means of a piezoelectric motor (Physik Instrumente). For quantification of TKS5 or FGD1 associated with invadopodia, three consecutive z-planes corresponding to the plasma membrane in contact with collagen fibers were projected using maximal-intensity projection in Fiji, and TKS5 or FGD1 signal was determined using the thresholding command excluding regions <8 px to avoid noninvadopodial structures. Surface covered by TKS5 or FGD1 signal was normalized to the total cell surface, and values were normalized to those of control cells.

### Linescan-based correlation of pixel fluorescence intensity of invadopodia markers
For quantification of the colocalization between the invadopodial markers the along collagen fibers, linescans were performed using Fiji software. Briefly, a segmented line (6-pixel width) was drawn along collagen fibers that were associated

with the invadopodial markers (such as TKS5, TKS4, cortactin, or FGD1). The fluorescence intensity signal of each pixel was measured, normalized to the maximal-intensity signal set to 100, and plotted on an x-y graph where the x axis represents the length of the invadopodia (in pixels) and the y axis represents the normalized fluorescence intensity of the markers. Correlation between the two markers was observed by plotting the normalized values of each invadopodial marker on a graph in which the x axis represents the normalized fluorescence intensity of one marker and the y axis represents the normalized fluorescence intensity of the other marker. Linear regression line was calculated using GraphPad Prism software.

## Quantification of Manders colocalization coefficient

Colocalization analyses were performed on Fiji software based on two tools (Schindelin et al., 2012). First, the Manders colocalization coefficient (Manders et al., 1993) gives a quantitative analysis of colocalization, since it indicates the proportion of one signal coinciding with the other signal, and is relatively insensitive to difference in signal intensities in both channels. Because thresholding is important by defining what is "significant" signal in both channels to compute the coefficient only on those pixels, it was performed manually on each image. The value of this coefficient was obtained using JaCOP plugin (Bolte and Cordelières, 2006). Second, we used the Plugin "Colocalization Colormap," encoding Jaskolski's algorithm (Jaskolski et al., 2005), which gives a spatial information about colocalization. For each pixel, a correlation between its value in one channel image and its value in the second channel image was computed based on normalized mean deviation product, which is represented on the collagen signal as a 2-color image (as in Fig. 4 E). This map enabled detection and comparison of the presence of colocalized pixels on the collagen fibers.

## Quantification of pericellular collagenolysis

Cells treated with indicated siRNAs were trypsinized and resuspended ($2.5 \times 10^5$ cells/ml) in 200 µl of ice-cold 2.0 mg/ml collagen I solution prepared as previously described. 40 µl of the cell suspension in collagen was added on a glass coverslip, and collagen polymerization was induced for 30 min by incubation at 37°C. L-15 complete medium was then added, and cells embedded in collagen were incubated for 16 h at 37°C in 1% $CO_2$. After fixation for 30 min at 37°C in 4% PFA in PBS, samples were incubated with anti-Col1-3/4C antibodies for 2 h at 4°C. After extensive washes, samples were counterstained with Cy3-conjugated anti-rabbit IgG antibodies and Phalloidin-Alexa Fluor 488 to visualize cell shape and mounted in DAPI. Image acquisition was performed with an A1R Nikon confocal microscope with a 40× NA 1.3 oil objective using a high-sensitivity 455-nm GaASP PMT detector and a 595 ± 50-nm bandpass filter. Quantification of degradation spots was performed as previously described (Monteiro et al., 2013). Briefly, maximal projection of 10 optical sections with 2-µm interval from confocal microscope z-stacks (20 µm depth) were preprocessed by a Laplacian-of-Gaussian filter using a homemade ImageJ macro 15. Detected spots were then counted and saved for visual verification. No manual correction was done. Degradation index was the number of degradation spots divided by the number of cells present in the field, normalized to the degradation index of control cells set to 100.

## Live-cell imaging in 3D type I collagen

Glass-bottom dishes (MatTek Corp.) were layered with 10 µl of a solution of 5 mg/ml unlabeled type I collagen mixed with 1/25 volume of Alexa Fluor 647–labeled collagen. Polymerization was induced at 37°C for 3 min as described above, and the bottom collagen layer was washed gently in PBS. 1 ml of cell suspension ($10^5$ cells/ml) in complete medium was added and incubated for 30 min at 37°C. Medium was gently removed, and two drops of a mix of Alexa Fluor 647–labeled type I collagen/unlabeled type I collagen at 2.0 mg/ml final concentration were added on top of the cells (top layer). After polymerization at 37°C for 90 min as described above, 2 ml of medium containing 20 ng/ml HGF was added to the culture. Z-stacks of images were acquired every 10 min during 16 h by confocal spinning disk microscopy (Roper Scientific) using a CSU22 Yokogawa head mounted on the lateral port of an inverted TE-2000U Nikon microscope equipped with a 40× 1.4-NA Plan-Apo objective lens and a dual-output laser launch, which included 491- and 561-nm, 50-mW DPSS lasers (Roper Scientific). Images were collected with a CoolSNAP HQ2 charge-coupled device camera (Roper Scientific). The system was steered by Metamorph 7 software. Kymographs were obtained with Fiji software along a line spanning the invadopodia diameter.

## Automated tracking of endosome angular distribution

A homemade Matlab program (available on demand) was developed to track nuclei based on nuclear staining and create a velocity-dependent coordinate system to analyze MT1-MMP endosomes relative to the direction of displacement of the nucleus (Infante et al., 2018). Nuclei were automatically segmented from maximal z-stack projection of sequential time frames (see previous section) based on smoothing and thresholding, and then were tracked based on the distance from their previous position. From the trajectory of each nucleus, speed (µm/min) and directionality (persistence) were computed for each consecutive pair of frames. A new polar coordinate system was defined such that the gravity center of the nucleus became the position (0,0) for all time points, and the velocity direction had an angle of 0°. This coordinate system then changed for each time point and was different for each nucleus. Endosomes around each nucleus were automatically segmented by Laplacian of Gaussian spot enhancement and marker-controlled watershed segmentation based on regional maxima. The coordinates of the positions of the center of gravity of all endosomes were then converted to this nucleus velocity-dependent coordinate system. Endosomes exactly in front of the nucleus in the direction of movement are then at 0°, and endosomes exactly at the rear of the displacement vector are at 180°. All data created for endosomes for all processed nuclei and all videos for one condition were then pooled to create a polar histogram (radar plot), showing the distribution of endosomes relative to the direction of nuclear movement.

### Inverted 3D collagen invasion assay

200 µl of 2.0 mg/ml collagen I was allowed to polymerize in Transwell inserts (Corning) for 2 h at 37°C as above. Cells were seeded on top of the collagen gel in complete medium, and 20 ng/ml HGF was added to the medium in the bottom chamber of the Transwell as chemoattractant. After 48 h of seeding, cells were fixed and stained with DAPI and visualized by confocal microscopy with serial optical sections captured at 10-µm intervals with a 10× objective on a Zeiss LSM510 confocal microscope. Invasion was measured by dividing the sum of DAPI signal intensity of all slides beyond 30 µm (invading cells) by the sum of the intensity of all slides (total cells).

### Statistics and reproducibility

All results were presented as mean ± SEM of three independent experiments. GraphPad Prism was used for statistical analysis. Statistical significance was defined as *, $P < 0.05$; **, $P < 0.01$; ***, $P < 0.001$; ****, $P < 0.0001$; and ns, not significant. Data were tested for normal distribution using the D'Agostino–Pearson normality test, and nonparametric tests were applied otherwise. One-way ANOVA, Kruskal–Wallis, and Mann–Whitney $U$ tests were applied as indicated in the figure legends.

### Online supplemental material

Fig. S1 shows additional evidence that TKS5 is required for invadopodia formation and function in collagenolysis. Fig. S2 documents the silencing of the proteins of interest upon siRNA treatment in MDA-MB-231 and Hs578T breast cancer cells. Fig. S3 shows the effect of FGD1 knockdown on TKS5-, cortactin-positive invadopodia formation and the consequence of TKS5, FGD1, or CDC42 silencing on collagenolysis by MDA-MB-231 cells in a 3D collagen gel. Fig. S4 shows that the integrity of the fifth SH3 domain of TKS5 is required for its coimmunoprecipitation with FGD1. Fig. S5 shows the absence of coimmunoprecipitation between TKS4GFP and FGD1, while coimmunoprecipitation is detected between TKS5GFP and FGD1, as well a comparison of the collagenolysis activity of MDA-MB-231 and Hs578T cells in a 3D collagen gel. Video 1 shows a representative example of MDA-MB-231 cells coexpressing FGD1GFP and TKS5mCherry plated on a fibrillary type collagen layer, documenting the dynamics of the double-positive invadopodia structures. Table S1 lists the proteins interacting with TKS5GFP identified by coimmunoprecipitation and MS analysis. The interactions of TKS5 with disintegrin and metalloproteinase domain–containing protein 15, monocarboxylate transporter 4, 14–3-3 family proteins, and FGD1 were previously reported (Abram et al., 2003; Couzens et al., 2013; Hein et al., 2015). Table S2 lists the antibodies used in this study. Table S3 lists the siRNAs and plasmids used in this study.

## Acknowledgments

We thank Drs. S. Courtneidge (Oregon Health & Science University, Portland, OR), L. Cantley (Weill Cornell Medical College, New York, NY), and C. Lamaze for providing reagents for this study, Sabrina Rousseau and T. Dakhli for cloning the GST-FGD1 fusion proteins, and Dr. A. El Marjou and K. Niort for the purification of the GST-FGD1 proteins.

We thank the Nikon Imaging Centre at Institut Curie-Centre National de la Recherche Scientifique and Cell and Tissue Imaging Facility of Institut Curie, member of the France Bio Imaging national research infrastructure (Agence Nationale de la Recherche, ANR-10-INBS-04), for help with image acquisition. A. Zagryazhskaya-Masson and A. Duperray-Susini were supported by a grant from Institut Curie (PIC3i InvadoROS) and a postdoctoral fellowship from Ligue Contre le Cancer (to A. Zagryazhskaya-Masson). A. Castagnino was supported by Worldwide Cancer Research (grant 16-1235). A.-S. Macé was supported by a grant from Labex (CelTisPhyBio ANR-11-LABX-0038), part of the IDEX PSL no. ANR-10-IDEX-0001-02. E. Infante was supported by a postdoctoral fellowship from Ligue Contre le Cancer. R. Ferrari was supported by a fellowship from Fondation ARC pour la Recherche sur le Cancer. This work was supported by grants from Ligue Contre le Cancer (Comité de la Gironde) to E. Génot, "Région Ile-de-France" and Fondation pour la Recherche Médicale to D. Loew, Fondation ARC pour la Recherche sur le Cancer (PGA1 RF20170205408), and core funding from Institut Curie and Centre National de la Recherche Scientifique to P. Chavrier.

The authors declare no competing financial interests.

Author contributions: A. Zagryazhskaya-Masson and P. Monteiro designed and performed most of the experiments and analyzed and interpreted the data. P. Monteiro helped P. Chavrier write the manuscript. A.-S. Macé wrote ImageJ macros for image analysis. A. Castagnino and R. Ferrari performed endosome tracking experiments and interpreted the data. E. Infante performed the inverted invasion assay and interpreted the data. A. Duperray-Susini and P. Monteiro conducted the molecular biology experiments necessary for the generation of the TKS5 constructs. F. Dingli generated and analyzed the MS data under D. Loew's supervision. A. Lanyi and E. Génot provided essential reagents for the study. E. Génot helped design and interpret some of the experiments. A. Zagryazhskaya-Masson, P. Monteiro, E. Infante, E. Génot, and P. Chavrier actively engaged in discussions throughout the study. A. Zagryazhskaya-Masson, E. Infante, and E. Génot edited the manuscript. P. Chavrier supervised the study, designed the experiments, interpreted the data, and wrote the manuscript.

**Note added in proof.** During the final revision of the present article, Thuault et al. (2020) identified an interaction between TKS5 and FGD1 using the BioID proximity biotinylation technology.

Submitted: 18 October 2019

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

# Supplemental material

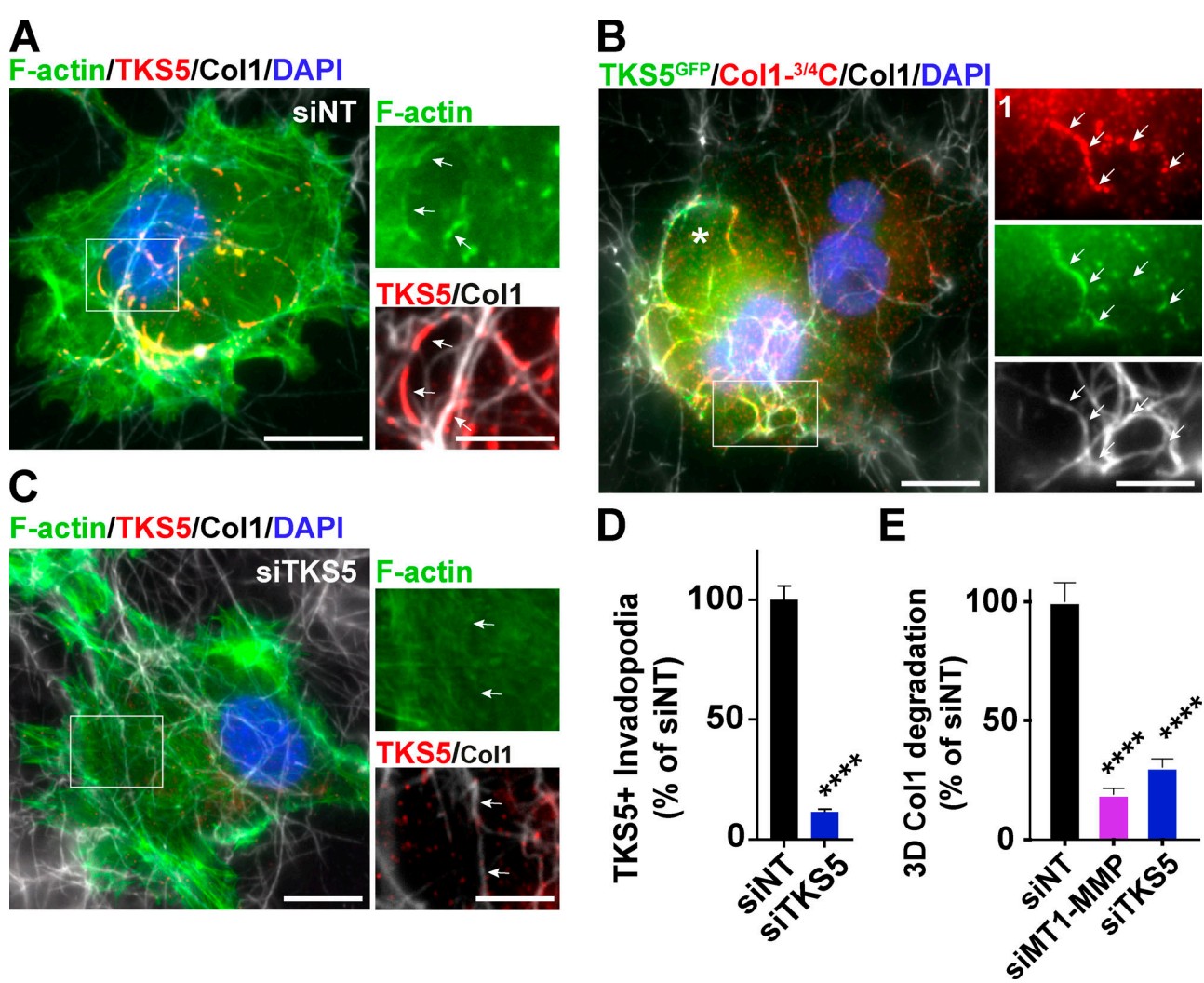

Figure S1.    **TKS5 is required for invadopodia formation and function. (A)** MDA-MB-231 cells were cultured on a fibrillary layer of type I collagen (gray) for 60 min and stained for TKS5 (red), F-actin (green), and DAPI (blue). **(B)** MDA-MB-231 cells expressing TKS5$^{GFP}$ (green) were cultured on a collagen layer (gray) for 60 min. Cleaved collagen was stained with the Col1-$^{3/4}$C antibody (red). Cell nuclei were stained with DAPI (blue). Scale bar, 10 µm; zoom-in of boxed region, 5 µm. **(C)** Cells depleted for TKS5 were analyzed as in A. **(D)** Quantification of TKS5-positive invadopodia in siRNA-treated MDA-MB-231 cells. The y axis is the ratio of the TKS5 area to the total cell area normalized to the mean value of control siNT-treated cells (as percentage ± SEM). siNT, $n$ = 73 cells; siTKS5, $n$ = 66 cells from three independent experiments. Data were analyzed using $t$ test. **(E)** Pericellular collagenolysis by the indicated MDA-MB-231 cell populations measured as mean intensity of Col1-$^{3/4}$C signal per cell ± SEM. Values for control cells were set to 100%. siNT, $n$ = 61 cells; siMT1-MMP, $n$ = 63 cells; and siTKS5, $n$ = 42 cells from three independent experiments. Kruskal–Wallis test. ****, $P < 0.0001$.

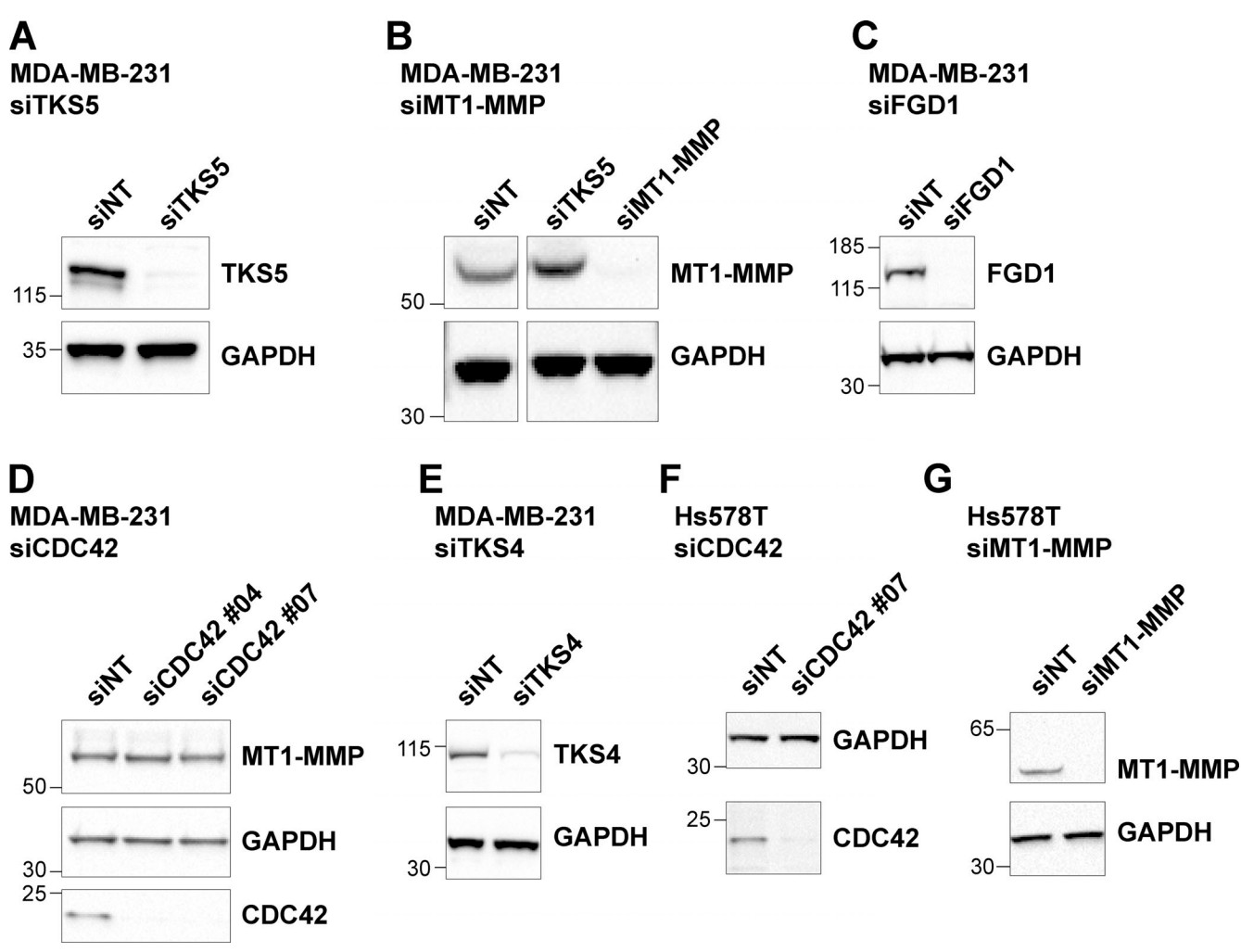

Figure S2. **Analysis of protein expression in siRNA-treated cells. (A–G)** MDA-MB-231 or Hs578T human breast cancer cells were treated with the indicated siRNAs for 72 h and analyzed by immunoblotting analysis with the indicated antibodies. Equal loading was controlled using GAPDH antibody. Molecular weights are in kD.

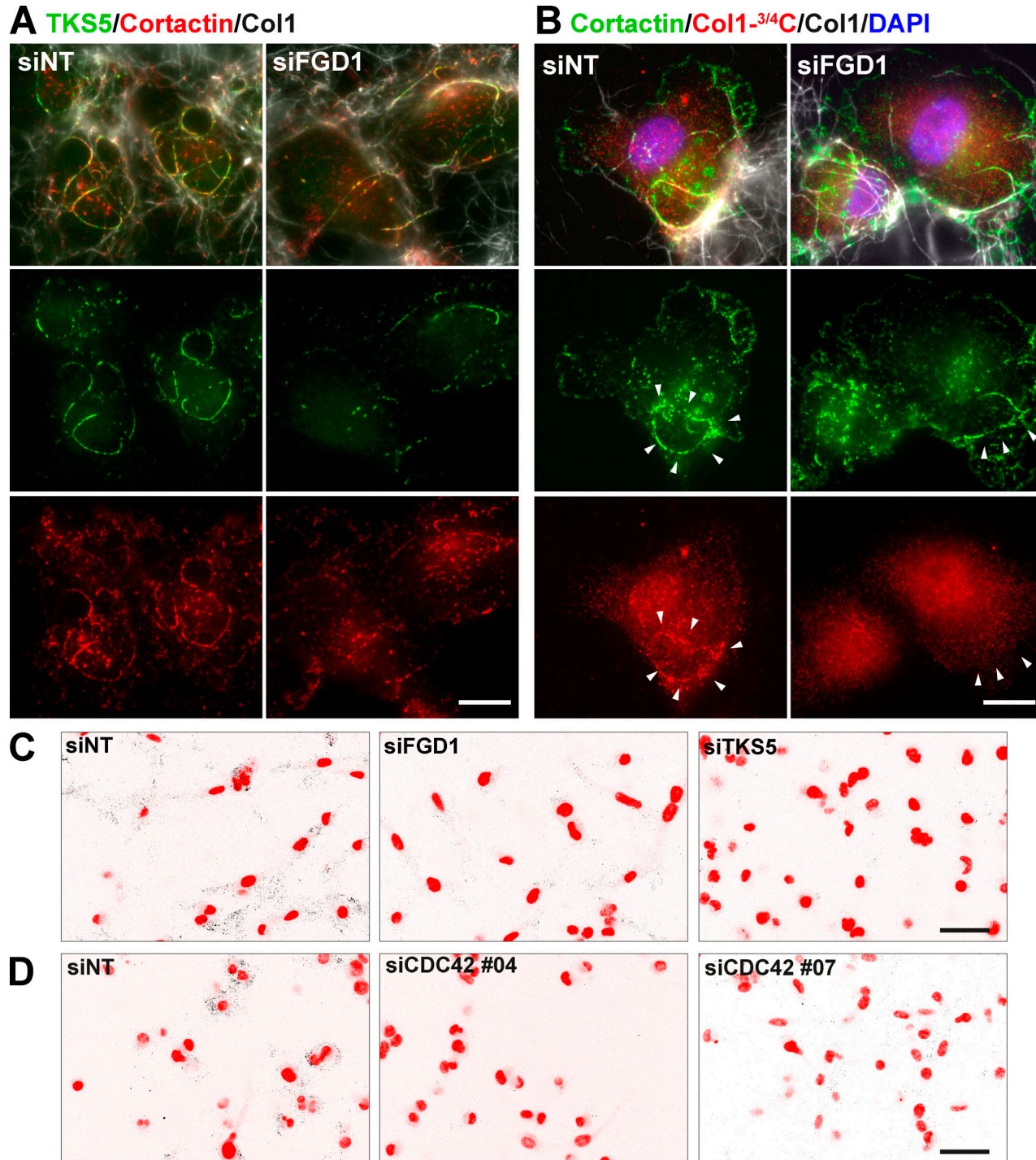

Figure S3.   **Pericellular collagen degradation requires the components of the TKS5/FGD1/CDC42 axis. (A)** MDA-MB-231 cells treated with FGD1 siRNA were cultured on a fibrillary layer of type I collagen (gray) and stained for TKS5 (green) and cortactin (red). **(B)** Cells as in B stained for cortactin (red). Cleaved collagen was stained with the Col1-3/4C antibody (red). Cell nuclei were stained with DAPI (blue). Scale bars, 10 µm. **(C and D)** Representative images of pericellular collagenolysis detected with Col1-3/4C antibody (black signal in the inverted images). Nuclei were stained with DAPI (pseudo-colored in red). Scale bar, 50 µm.

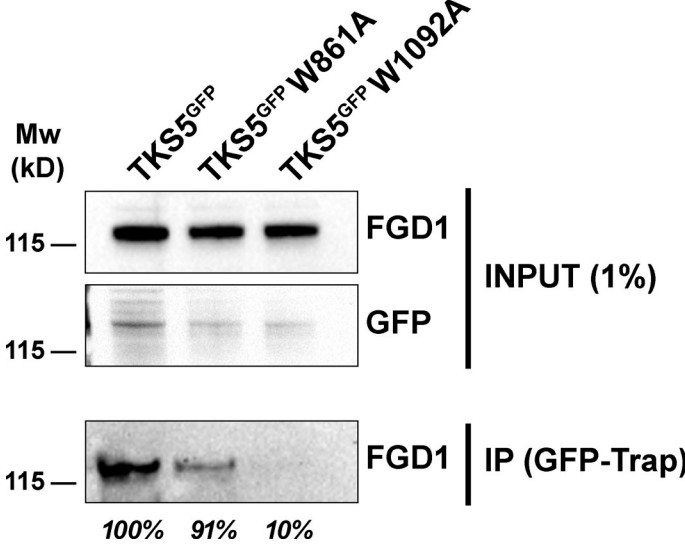

Figure S4. **The fifth SH3 domain of TKS5 is required for binding to FGD1.** Lysates of MDA-MB-231 cells expressing the indicated GFP-tagged constructs were immunoprecipitated with anti-GFP nanobody. Bound proteins were analyzed with FGD1 antibodies (IP). 1% of total lysate was loaded as a control (input). The binding capacity of the different TKS5 constructs to FGD1 is indicated in percentage of wild-type TKS5.

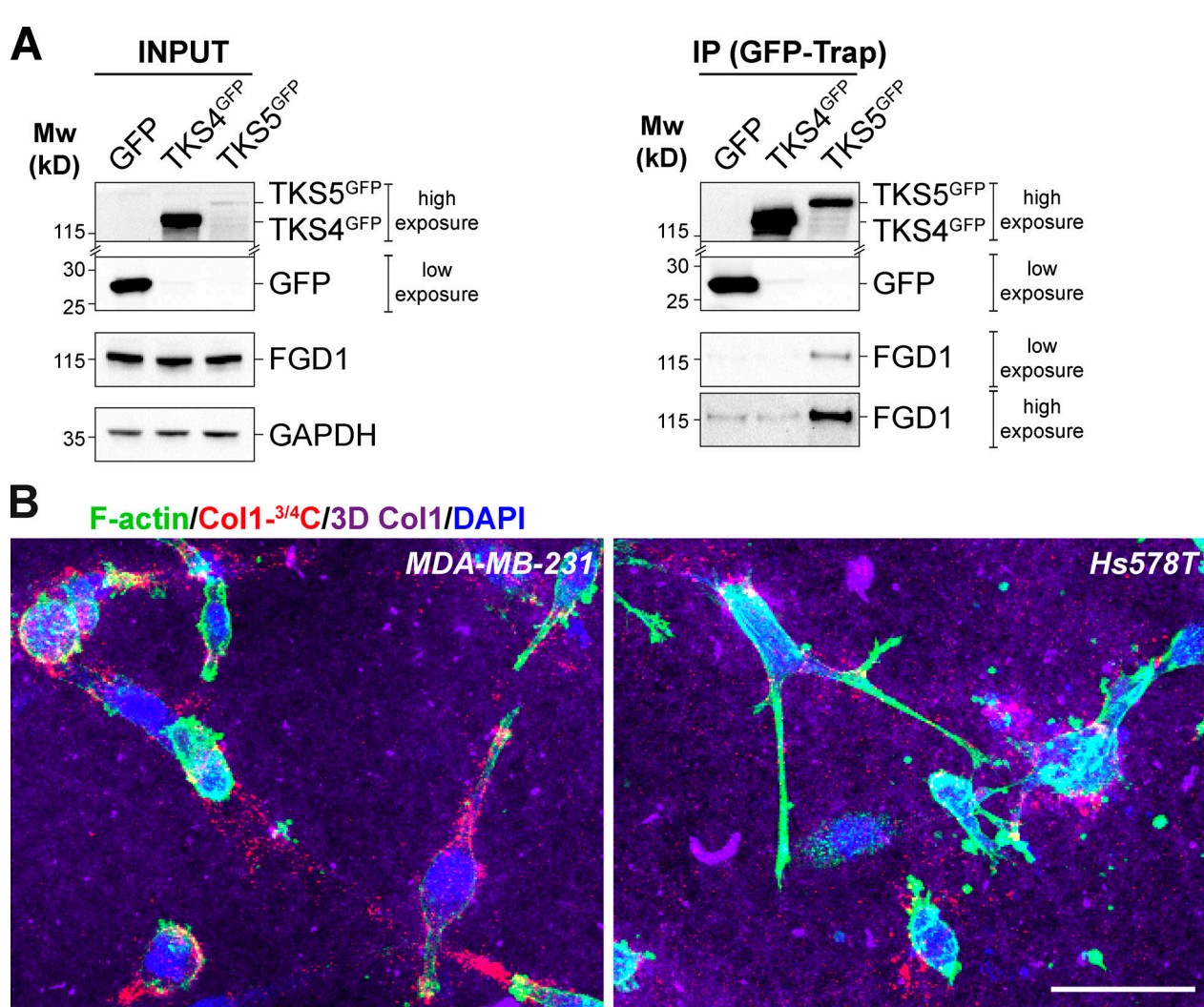

**Figure S5.  TKS4 does not interact with FGD1. (A)** Lysates of MDA-MB-231 cells expressing TKS4^GFP, TKS5^GFP, or GFP (Ctrl) were immunoprecipitated with GFP antibodies. Bound proteins were analyzed with TKS4, TKS5, GFP, and FGD1 antibodies (GFP-Trap). Note that TKS4^GFP and TKS5^GFP signals correspond to a longer exposure than that of the GFP signal. In addition, in the immunoprecipitated materials (GFP-Trap), two different exposure times are shown for the FGD1 signal, indicating that there is no detectable binding of FGD1 to TKS4^GFP. 5% of total lysates was loaded as a control (input). Equal loading was controlled using GAPDH antibody. Molecular weights are in kD. **(B)** MDA-MB-231 or Hs578T cells were embedded in 3D type I collagen gel (magenta) for 16 h and then stained for cytoskeletal F-actin (green). Pericellular collagenolysis was detected with the Col1-^3/4C antibody (red). Nuclei were stained with DAPI (blue). Scale bar, 50 µm.

**Video 1.  TKS5 and FGD1 colocalize in dynamic collagen-remodeling invadopodia.** Left panel: MDA-MB-231 cells expressing FGD1^GFP (green) and TKS5^mCherry (red) were plated on top of a thin type I collagen layer (gray). A time-lapse sequence was started after 45 min by confocal spinning-disk microscopy. Images were taken every min during 45 min. The right panel shows the active remodeling of the collagen fibers by the TKS5 FGD1 double-positive cell that can be compared with collagen remodeling by two nontransfected cells visible on the upper and lower right corners. Frame rate of the video is 6 frames/s.

**Three tables are provided online. Table S1 lists proteins corresponding to the peptides quantified by LC-MSMS in GFP-Trap immunoprecipitates from TKS5GFP-expressing MDA-MB-231 cells versus wild-type MDA-MB-231 cells. Table S2 lists antibodies used in this study. Table S3 lists siRNAs and plasmids used in this study.**

