## [Peer Review File · The Journal of Cell Biology]

Intersection of TKS5 and FGD1/CDC42 signaling cascades directs the formation of invadopodia

Anna Zagryazhskaya-Masson, Pedro Monteiro, Anne-Sophie Macé, Alessia CASTAGNINO, Robin Ferrari, Elvira Infante, Aléria Duperray-Susini, Florent Dingli, Árpád Lányi, Damarys Loew, Elisabeth Genot, and Philippe Chavrier

Corresponding Author(s): Philippe Chavrier, Institut Curie and Pedro Monteiro, Institut Curie

Review Timeline:	Submission Date:	2019-10-18
	Editorial Decision:	2019-11-15
	Revision Received:	2020-04-24
	Editorial Decision:	2020-05-22
	Revision Received:	2020-05-28

Monitoring Editor: Kenneth Yamada

Scientific Editor: Melina Casadio

Transaction Report:

DOI: <https://doi.org/10.1083/jcb.201910132>

November 15, 2019

Re: JCB manuscript #201910132

Dr. Philippe Chavrier
Institut Curie
Institut Curie - Section Recherche CNRS UMR144 26 rue d'Ulm
Paris 75248
France

Dear Dr. Chavrier,

Thank you for submitting your manuscript entitled "Intersection of TKS5 and FGD1/CDC42 signaling cascades directs the formation of collagenolytic invadopodia". The manuscript was assessed by expert reviewers, whose comments are appended to this letter. As you can see from the expert reviews from three top leaders in the fields spanned by your potentially important paper, the findings in your manuscript were considered interesting and potentially appropriate for JCB, but only if some major issues could be resolved requiring further in-depth analyses. Interestingly, their reviews echoed some key points raised at the initial editorial evaluation stage, where it was felt that more in-depth mechanistic insight and more than one cell type seemed to be needed - but we felt that it deserved full peer review in the hope of identifying specific experimental points from leading experts in the field that could lead to a superior study. We invite you to submit a revision if you can address the reviewers' key concerns, as outlined here.

One key point raised involves the need for further characterization of the interaction between TKS5 and FGD1 in terms of whether they are directly rather than indirectly interacting in a linear pathway. The reviewers provide constructive suggestions about how to dissect this interaction by mutagenesis and a suggested double knockdown. Some type of approach examining the involvement of CDC42 would also help considerably. There would also need to be resolution of concerns about the strength of the evidence PI3,4P vs. PI4,5P data with clearer data and attempted immunofluorescence staining of phosphoinositides. In addition, this study would need as much quantification as is practical, as well as ensuring the use of 2-3 independent siRNAs and/or knockdown siRNA-resistant rescue experiments. Please also check whether the key conclusions can be found in another cell type. We would like to leave it up to you and your colleagues about whether to modify the second part of the study and hope that the other insightful, constructive points raised by these expert reviewers can be addressed. We very much hope that you can return a manuscript that resolves these issues within the 3-4 month JCB resubmission period, but if it takes longer, we would be open to an extension - though there would first need to be a re-examination of the novelty of these findings (the latter is not needed for a < 4-month resubmission).

We look forward to receiving a manuscript revised as indicated above, which would be returned to the three conscientious reviewers for their final evaluations and input regarding appropriateness for publication. We sincerely thank you for submitting this intriguing study to JCB.

While you are revising your manuscript, please also attend to the following editorial points to help expedite the publication of your manuscript. Please direct any editorial questions to the journal

office.

GENERAL GUIDELINES:

Text limits: Character count for an Article is < 40,000, not including spaces. Count includes title page, abstract, introduction, results, discussion, acknowledgments, and figure legends. Count does not include materials and methods, references, tables, or supplemental legends.

Figures: Articles may have up to 10 main text figures. Figures must be prepared according to the policies outlined in our Instructions to Authors, under Data Presentation, <http://jcb.rupress.org/site/misc/ifora.xhtml>. All figures in accepted manuscripts will be screened prior to publication.

*****IMPORTANT:** It is JCB policy that if requested, original data images must be made available. Failure to provide original images upon request will result in unavoidable delays in publication. Please ensure that you have access to all original microscopy and blot data images before submitting your revision.*******

Supplemental information: There are strict limits on the allowable amount of supplemental data. Articles may have up to 5 supplemental figures. Up to 10 supplemental videos or flash animations are allowed. A summary of all supplemental material should appear at the end of the Materials and methods section.

The typical timeframe for revisions is three months; if submitted within this timeframe, novelty will not be reassessed at the final decision. Please note that papers are generally considered through only one revision cycle, so any revised manuscript will likely be either accepted or rejected.

Thank you for this interesting contribution to the Journal of Cell Biology. You can contact us at the journal office with any questions, cellbio@rockefeller.edu or call (212) 327-8588.

With kind regards,

Ken Yamada

Kenneth Yamada, MD, PhD
Editor, Journal of Cell Biology

Melina Casadio, PhD
Senior Scientific Editor, Journal of Cell Biology

Reviewer #1 (Comments to the Authors (Required)):

In this work, the authors report the identification of the CDC42 GEF as an interactor of the invadopodia key regulator TKS5. They further propose that a TKS5-FGD1-CDC42-IQGAP axis operates in certain cells to control linear convex invadopodia formation, collagen degradation activity, and MT1-MMP polarized distribution in cell migrating on top of reconstituted ECM. They further show that contrary to TKS5, TKS4 does not associate with FGD1 and suggests the existence of context (cell)-dependent, diverse pathways necessary to control collagen degradation and invasiveness.

As usual, the quality of the work from this group is excellent and particularly notable is the imaging in support of some of the conclusions of this manuscript. Less exciting and compelling is the set of experiments in support of the existence of a TKS5-FGD1-CDC42-IQGAP axis in the control of MT1-MMP dynamics, activity and collagenolytic activity. Firstly, the fact that individual silencing of each of this protein impairs matrix degradation and the number of TKS5+ linear structures is not a demonstration that they operate as a linear pathway. Each of the components could potentially act independently. One would have expected a more detailed set of experiments exploiting, for example, TKS5 mutants in the SH3 domains 4 and 5, that are shown to mediate the interaction with FGD1, or the corresponding mutant in proline-rich region in FGD1 to provide a direct and more compelling evidence that at least the association of these two proteins is indeed essential for linear invadopodia formation and activity. The authors could also exploit the structural/functional difference between TKS4 and TKS5. All these experiments would ideally require the reconstitution of siRNA treated cells with WT and mutant proteins (at least the use of multiple oligos should be included for all the siRNA experiments-).

An alternative/complementary experiment that is predicted based on the newly identified association between TKS5 and FGD1 is that the local activation of CDC42 should be dramatically impaired. While performing FRET experiments might be challenging, they would greatly corroborate the linear model proposed and the key functional role of FGD1. Testing whether fast cycling or possibly constitutive active CDC42 mutants bypass the requirement for TKS5-FGD1 in the formation of linear invadopodia and matrix degradation would be another possible avenue worth exploring

There are also other possibly minor issues to be addressed.

1. The results presented in Figure 1 are neat and clear-cut, but entirely confirmatory of previous findings and long-established knowledge. This figure could be reduced to the identification of TKS5 in linear invadopodia as it represents a very marginal advance over previous findings.
2. In figure 2, it is shown that PI3,4P rather than PI4,5P specifically accumulates a curved membrane of invadopodia. However, some cautions should be exerted in interpreting this set of findings: Firstly, as stated by the authors PM associated probes would appear apparently enriched at these PM folded sites. The difference between the intensity distribution of Tubby-GFP that recognizes PI4,5P and TAPP1-GFPm, that binds to PI3,4P is not that striking. Additionally, the enrichment along of linear invadopodia of INPP4b that converts PI3,4P into PI3P would suggest that these structures should have reduced levels of PI3,4P and increased PI3P. Is this the case? Somewhat less straightforward is also the distribution of p130Cas used here as a proxy for the accumulation of the 5'-inositol phosphatase SHIP2. Firstly, it should be justified why SHIP2 itself was not examined as opposed to p130Cas. Secondly, the images in 2E do not really show a sharp increase in p130Cas signal at the convex invadopodial PM sites. Hence, the conclusion of this set of experiments appears overstated.
3. On the Mass spectrometry results: it would seem necessary to deposit the raw data from which the enrichment ratio has been calculated in repository sites along with the Mass Spect chromatograms.
4. "FGD1 knockdown (Supplementary Figure 1C), induced a partial significant ~35 % reduction of TKS5-positive invadopodia assembly (Figure 4E and Supplementary Figure 2A)."
I guess the author intended a partial, BUT significant rather than a partial significant down-

regulation.

In this experiment, more than 1 oligo against FGD1 and reconstitution experiments also appear necessary to claim a specific involvement of FGD1 in invadopodia formation.

Reviewer #2 (Comments to the Authors (Required)):

This study by Zagrayazhskaya-Masson and colleagues highlights an interesting interaction of invadopodial proteins TKS5 with FGD, a CDC42 GEF, at invadopodia in a breast cancer cell line. It implicates specific PIPs and downstream effectors of this complex with the localization and action of the MT1-MMP protease in cells plated on type 1 collagen. The study makes some interesting findings although it comes across as diffuse and over ambitious overall.

Specific suggestions that seem important to clarify are:

- It is unclear how this interaction was first identified and by whom. Was it first reported by Hess and colleagues using a Mann-like screen and this current group performed the needed characterization and cell biological analysis? So perhaps this interaction was already known?

-Are the reported interactions direct or indirect? It appears that all the GST or GFP trap pulldowns were conducted in cell lysates so is there an adaptor mediating these interactions?

-This study does KDs of the putative interactors that give similar phenotypes which is encouraging, but is KD re-expression of the interactive mutants performed and compared in function to wt rescued cells? The specific cause and effects of disrupting the complex needs more experimentation.

-Is everything done in one cell type? Could several, but not all, of the key findings be performed in a second or third cell type to show increased relevance?

-This study provides insights into the type of PIP that is enriched at the invadopodial sites but does it provide functional insights into whether one PIP enhances recruitment and function of the proteins of interest to the degradation site or is this finding strictly observational?

The second part of the study linking the protein complex to other effectors and lysosome distribution and matrix degradation make the story more diffuse and perhaps overdone. It's not clear if the players involved contribute directly to these processes or not. Pursuit of the suggestions above applied to these other observations would help, or perhaps the overall story is too ambitious /complex and could be pared down.

Reviewer #3 (Comments to the Authors (Required)):

In this paper, the authors set out to characterize the contribution and mechanism of TKS proteins in the formation of collagenolytic invadopodia in breast cancer cells. They have identified the interaction between TKS5 and FGD1 at the collagenolytic invadopodia and mapped down their interacting domains. They further unveiled this signaling pathway regulating collagenolytic invadopodia through CDC42 and IQGAP1. The experiments are very well-designed, and all the data

strongly support the conclusions. With the improvement in quantification and certain extra experiments for validation, this paper should broaden the understanding of collagenolytic invadopodia formation.

The detail comments are as follows:

Whenever using the term of association/accumulation/colocalization in IF images, the authors are encouraged to quantify the colocalization in multiple cells. Showing just one cropped region of one cell without any quantitative analysis is not convincing.

Figure 1A-B, quantification for the colocalization of F-actin/cortactin with Col1 is needed for assessing their association.

The authors have concluded that the silencing of TKS5 strongly reduced the formation of F-actin-positive invadopodia (Figure 1B and Supplementary Figure 1A). Quantification of the number of invadopodia in the control and TKS5 KD cells is required for drawing this conclusion.

Figure 1C, quantification for the colocalization of TKS5GFP with Col1-3/4C is needed for assessing their association.

Figure 1G-H, it would be better to add a double KD of MT1-MMP and TKS5 to determine whether they have additive functions in cell invasion or not. This would be important to determine whether they are in one pathway as suggested by the model figure 7.

The PI(4,5)P2 and PI(3,4)P2 biosensor data are not convincing. Figure 2A, first, the Col1 pattern is completely different from Figure 1A. Secondly, it is not clear what the big chunk of TubbyGFP (ideally should be PIP2) cropped by the authors is, which is used for demonstrating the association of PIP2 with TKS5 and Col1. It is more than 30 μm in size. It may be just a cluster of TubbyGFP due to overexpression. Also, it is doubtful that the authors raised a statement that these data indicated homogenous distribution of PI(4,5)P2 at the plasma membrane. As a secondary messenger, PI(4,5)P2 is also enriched in the signaling hotspot instead of being homogeneously distributed. Most importantly, TubbyGFP could not differentiate the real presence of PIP2 (TubbyGFP-PIP2 in the complex) or TubbyGFP alone. There is a PIP2 antibody for immunofluorescent staining (Z-P045, Echelon Biosciences). This experiment should be confirmed by immunofluorescent staining of PIP2. Beyond endogenous PI(4,5)P2 staining, the use of PI(3,4)P2 and PI(3,4,5)P3 antibodies (also available from Echelon Biosciences) to stain endogenous phosphoinositides at the invadopodia could further strengthen the authors conclusion.

Figure 2B, the uncropped images are required for showing how the cell mask staining works for the whole cell.

Figure 2C, poor quality image for the TAPP1GFP (PI(3,4)P2 biosensor). It is not clear if this is the real signal or noises cross-activated by other channels. Also, what the single white arrow-head in the green channel is pointing at? This may not be aligned properly, and the corresponding red channel is missing. Again, there is a statement of association of PI(3,4)P2 with TKS5/Col1 without quantification.

Figure 2D, no quantification of colocalization of INPP4B with cortactin/Col1 for stating accumulation. Figure 2E, no quantification of colocalization of p130cas with MT1-MMP/Col1 for stating colocalization.

Figure 3C-D, the input for GST proteins and IB blot for the IP-ed GST proteins are missing. Also, it is

important to have the quantification.

Figure 4A-C, quantification is needed to conclude 'extensive co-localization'.

Figure 4G, no quantification for stating accumulation/association.

Figure 6B,E,D, no quantifications for stating association/co-localization.

Minor points:

1. *In vitro* and *in vivo* in the text should be italicized.

2. Figure 1G, no color code is labeled for the green staining.

3. There are two silQGAP1 #01 in Table S3. One should be silQGAP1 #03 according to Figure 5K, but this needs clarification.

4. In the abstract "Here, using co-immunoprecipitation experiments, we identify a direct interaction between TKS5 and FGD1, which is required for the assembly and function of collagenolytic invadopodium." can be changed to "Here, using co-immunoprecipitation and *in vitro* pulldown experiment's, we identify a direct interaction ..."

Rebuttal letter

JCB manuscript #201910132

Reviewer #1

We are grateful that the reviewer found our manuscript novel and interesting. We are also grateful for his constructive suggestions, which have helped to strengthen the story.

1. Firstly, the fact that individual silencing of each of this protein impairs matrix degradation and the number of TKS5+ linear structures is not a demonstration that they operate as a linear pathway. Each of the components could potentially act independently.

One would have expected a more detailed set of experiments exploiting, for example, TKS5 mutants in the SH3 domains 4 and 5, that are shown to mediate the interaction with FGD1, or the corresponding mutant in proline-rich region in association of these two proteins is indeed essential for linear invadopodia formation and activity.

As suggested by this Reviewer and by Reviewer#2 and #3, we analyzed the capacity of wild-type TKS5 and SH3#4 and SH3#5 domain mutants of TKS5 harboring a W-to-A substitution in a conserved residue critical for binding to Pro-rich motifs to colocalize with the endogenous FGD1 protein and to promote the formation of FGD1-positive, collagen fiber- associated structures in MDA-MB-231 cells silenced for the endogenous TKS5 protein. Co-immunoprecipitation assays demonstrated that the W1092A mutation in the SH3#5 domain strongly impaired FGD1 binding to 10% of FGD1 binding to wild-type TKS5, while binding was only slightly diminished by the analogous mutation in the SH3#4 domain (W861A, Supplementary Figure 4).

While the extent of TKS5/FGD1-positive invadopodia was similar in cells silenced for endogenous TKS5 and overexpressing WT or the W861A (SH3#4) mutant of TKS5, it was strongly reduced when cells expressed the W1092A (SH3#5) mutant of TKS5 (or a double mutant W861A/ W1092A), indicating that the integrity of the SH3#5 FGD1-binding domain of TKS5 was required for the capacity of TKS5 to rescue invadopodia formation (Figure 4D, E and G). In addition, quantification of a Mander's Overlap Coefficient as a measure of the colocalization between FGD1 and the overexpressed TKS5 constructs revealed that the integrity of TKS5 SH3#5 domain was similarly required for the colocalization of TKS5 and FGD1 in association with the collagen fibers (Figure 4E and F).

In reciprocal experiments, we investigated whether the overexpressed wild-type FGD1^{GFP} or a truncated mutant form of FGD1 with a deletion of the amino-

terminal TKS5-binding domain Pro-rich domain (PRD domain) could rescue the loss of endogenous FGD1 for the formation of TKS5-positive invadopodia in MDA-MB-231 cells treated with the FGD1 siRNA. Overexpressed wild-type FGD1 rescued invadopodia formation in FGD1-depleted cells. The deletion of the PRD domain of FGD1 impaired invadopodia formation (Figure 4HI). All together, these new data indicate that TKS5 and FGD1 work in a linear pathway and that their interaction is required for invadopodia formation in MDA-MB-231 cells.

- The authors could also exploit the structural/functional difference between TKS4 and TKS5. All these experiments would ideally require the reconstitution of siRNA treated cells with WT and mutant proteins (at least the use of multiple oligos should be included for all siRNA experiments).

In addition to the mix of four independent siRNAs that we used in the initial version of the manuscript (Smartpool), we have now added additional independent siRNAs to target the 3'UTR sequence of the TKS5 or FGD1 transcripts. In addition, two independent siRNAs have been used to target the CDC42 protein.

- An alternative/complementary experiment that is predicted based on the newly identified association between TKS5 and FGD1 is that the local activation of CDC42 should be dramatically impaired. While performing FRET experiments might be challenging, they would greatly corroborate the linear model proposed and the key functional role of FGD1.

Testing whether fast cycling or possibly constitutive active CDC42 mutants bypass the requirement for TKS5-FGD1 in the formation of linear invadopodia and matrix degradation would be another possible avenue worth exploring

As suggested by this referee, we examined the consequences of expressing GFP-tagged CDC42 constructs in MDA-MB-231 cells by live-cell confocal spinning disk microscopy. The GFP signal for wild-type CDC42 was homogeneously and diffusely distributed in the plasma membrane with a brighter signal in association with the collagen fibers and in TKS5mCherry-positive structures (see Figure A appended at the end of this rebuttal letter for the Referees only). In addition, we observed that this pattern was strictly identical to the distribution of CellMask, a lipophilic dye that homogeneously stains the plasma membrane (not shown). GFP-tagged CDC42-expressing cells were able to remodel the underneath collagen fibers similar to non-transfected cells (Appended Figure B and D). Our conclusion is that overexpressed wild-type CDC42 associates with the plasma membrane probably due to its isoprenylated carboxy-terminal domain but there is no indication of a specific enrichment in invadopodia. Additionally, we analyzed the distribution of the

constitutively active GTPase-defective CDC42-V12 mutant. In this case, we observed an enrichment of CDC42-V12 in association with the collagen fibers (Appended Figure C). However, CDC42-V12 recruitment did not seem to be correlated with an increased remodeling of the collagen fibers; rather, the collagen fibers underneath cells overexpressing GFP-CDC42-V12 tended to be less remodeled than untransfected or GFP-CDC42 WT-expressing cells (Appended Figure D). At this stage, we would like to suggest that overexpression of CDC42 may not be optimal to characterize the effect of CDC42 at a precise location inside the cell. In addition, collagen remodeling may require the capacity of CDC42 to hydrolyse GTP and cycle between the GDP and GTP-bound conformations. In addition, FGD1, which is bypassed by constitutively active CDC42-V12 may have non-catalytic function(s) such as the activation of Arp2/3 complex actin assembly by interacting with cortactin (Kim et al. 2004).

Therefore, these preliminary experiments indicate that it will be of interest to analyze further the dynamics of CDC42 in collagenolytic invadopodia as suggested by this Referee. Future cutting-edge experiments (i.e. using optogenetic, FRET-based tools as suggested by this referee and possibly CRISPR/CAS9 gene-editing to achieve endogenous expression of the tagged proteins), beyond the scope of the present study, will be necessary to decipher burning issues such as the spatially-controlled dynamics and function of CDC42 in collagenolytic invadopodia.

There are also other possibly minor issues to be addressed

1. The results presented in Figure 1 are neat and clear-cut, but entirely confirmatory of previous findings and long-established knowledge. This figure could be reduced to the identification of TKS5 in linear invadopodia as it represents a very marginal advance over previous findings.

We agree. Figure 1 in the revised manuscript has been reduced to the identification of TKS5 in linear invadopodia.

2. In figure 2, it is shown that PI3,4P rather than PI4,5P specifically accumulates a curved membrane of invadopodia. However, some cautions should be exerted in interpreting this set of findings: Firstly, as stated by the authors PM associated probes would appear apparently enriched at these PM folded sites. The difference between the intensity distribution of Tubby-GFP that recognizes PI4,5P and TAPP1-GFPm, that binds to PI3,4P is not that striking.

In order to strengthen PI4,5P2 and PI3,4P2 distribution data and in response to related comments and suggestions by Reviewer #2 and #3, we purchased the

Z-P045 anti-PI(4,5)P2 antibodies and Z-P034 anti-PI(3,4)P2 antibodies from Echelon Biosciences and stained MDA-MB-231 cells plated for 60 min on collagen fibers. Cells were counterstained for cortactin to label invadopodia. Unfortunately, in our hands, labeling looked unspecific and staining obtained with these two antibodies was not convincing. As we were not able to confirm and consolidate the phosphoinositide distribution data that we previously generated using genetically encoded PI(4,5)P2 and PI(3,4)P2 sensors, Tubby and TAPP1, respectively, we have decided to remove Tubby and TAPP1-based dataset in the revised manuscript.

the enrichment along of linear invadopodia of INPP4b that converts PI3,4P into PI3P would suggest that these structures should have reduced levels of PI3,4P and increased PI3P. Is this the case?

The initial and main purpose of this experiment was to use overexpressed FLAG-tagged INPP4B as a PI(3,4)P2 sensor based on INPP4B substrate affinity. Immunofluorescence detection of FLAG-tagged INPP4B and counterstaining for invadopodial TKS5 and quantification of the correlation of FLAG-tagged INPP4B and TKS5 pixel fluorescence intensity along elongated invadopodia are reported in Figure 7EF. Our data indicate that FLAG-tagged INPP4B can be detected in TKS5-positive invadopodia suggesting some accumulation of PI(3,4)P2 substrate.

We agree that accumulation of INPP4B^{FLAG} in invadopodia may reduce PI(3,4)P2 and increase PI3P levels in these structures. However, it is worth considering early-on data from the group of S. Courtneidge who found that the PX domain of TKS5 has affinity for both PI3P and PI(3,4)P2 (Abram et al., 2003). Thus, we believe it may be difficult to predict the effect of PI(3,4)P2 conversion to PI3P catalyzed by INPP4B as far as TKS5 recruitment or function are concerned.

Somewhat less straightforward is also the distribution of p130Cas used here as a proxy for the accumulation of the 5'-inositol phosphatase SHIP2. Firstly, it should be justified why SHIP2 itself was not examined as opposed to p130Cas. Secondly, the images in 2E do not really show a sharp increase in p130Cas signal at the convex invadopodial PM sites. Hence, the conclusion of this set of experiments appears overstated.

As suggested by this referee, we have performed a new set of immunofluorescence analyses to address the distribution of endogenous SHIP2 and p130CAS in relation with cortactin. New images and quantification are reported in Figure 7A-C in the revised manuscript.

3. On the Mass spectrometry results: it would seem necessary to deposit the raw data from which the enrichment ratio has been calculated in repository sites along with the Mass Spect chromatograms.

The mass spectrometry proteomics data have been deposited to the ProteomeXchange Consortium via the PRIDE partner repository with the dataset identifier PXD011632 (reviewer55167@ebi.ac.uk, EO2jNnfD). This information is provided in the Material and Methods section (Mass spectrometry analysis).

4. "FGD1 knockdown (Supplementary Figure 1C), induced a partial significant ~35 % reduction of TKS5-positive invadopodia assembly (Figure 4E and Supplementary Figure 2A)."

The sentence has been reworded.

In this experiment, more than 1 oligo against FGD1 and reconstitution experiments also appear necessary to claim a specific involvement of FGD1 in invadopodia formation.

In addition to the mix of four independent FGD1-specific siRNAs (Smartpool) that we initially used in the first version of the manuscript, we are now reporting new experiments using a mix of two additional independent siRNAs to target the 3'UTR of the FGD1 transcript. The requested rescue experiments have been performed showing that wild-type, but not a truncated form of FGD1 deleted of its amino-terminal PRD domain, can rescue the formation of TKS5-positive invadopodia in FGD1-depleted cells (see Figure 4HI).

Reviewer #2

This study by Zagrayazhskaya-Masson and colleagues highlights an interesting interaction of invadopodial proteins TKS5 with FGD, a CDC42 GEF, at invadopodia in a breast cancer cell line. It implicates specific PIPs and downstream effectors of this complex with the localization and action of the MT1-MMP protease in cells plated on type 1 collagen. The study makes some interesting findings although it comes across as diffuse and over ambitious overall.

We thank this referee for his/her constructive criticisms and helpful suggestions.

Specific suggestions that seem important to clarify are:

- It is unclear how this interaction was first identified and by whom. Was it first reported by Hess and colleagues using a Mann-like screen and this current group performed the needed characterization and cell biological analysis? So perhaps this interaction was already known?

We are aware about one study from Matthias Mann's laboratory describing a global human interactome study, in which an interaction between TKS5 and FGD1 has been reported (Hein et al., Cell. 163:712-723, 2015) but was not further validated. This study is cited in our manuscript.

- Are the reported interactions direct or indirect? It appears that all the GST or GFP trap pulldowns were conducted in cell lysates so is there an adaptor mediating these interactions?

We are describing new experiments aimed at a better mapping the FGD1-interaction domain of TKS5 based on point mutations in the fourth or fifth SH3 domain of TKS5. Co-immunoprecipitation experiments showed that the integrity of the fifth SH3 domain of TKS5 is required for binding – directly or indirectly - to FGD1 (see supplemental Figure 4).

Although we agree that the issue as to whether the interaction between FGD1 and TKS5 is direct or not is of interest, biochemical assessment of this interaction would require recombinant proteins that were not easily available to us. Instead, we found that the fifth SH3 domain of TKS5 is required for interaction with the PRD-domain of FGD1, and thus it is very likely that the interaction between these two proteins is direct and based on well-known SH3 domain-based interaction with Proline-rich motifs in the target – FGD1 – protein.

- This study does KDs of the putative interactors that give similar phenotypes which is encouraging, but is KD re-expression of the interactive mutants performed and compared in function to wt rescued cells? The specific cause and effects of disrupting the complex needs more experimentation.

As suggested by this Reviewer and by Reviewer#1 and #3, we analyzed the capacity of wild-type TKS5 and SH3#4 and SH3#5 domain mutants of TKS5 harboring a W-to-A substitution in a conserved residue critical for binding to Pro-rich motifs to colocalize with the endogenous FGD1 protein and to promote the formation of FGD1-positive, collagen fiber- associated structures in MDA-MB-231 cells silenced for the endogenous TKS5 protein. Co-immunoprecipitation assays demonstrated that the W1092A mutation in the SH3#5 domain strongly impaired FGD1 binding to 10% of FGD1 binding to wild-type TKS5, while binding was only slightly diminished by the analogous mutation in the SH3#4 domain (W861A, Supplementary Figure 4).

While the extent of TKS5/FGD1-positive invadopodia was similar in cells silenced for endogenous TKS5 and overexpressing WT or the W861A (SH3#4) mutant of TKS5, it was strongly reduced when cells expressed the W1092A (SH3#5) mutant of TKS5 (or a double mutant W861A/ W1092A), indicating that the integrity of the SH3#5 FGD1-binding domain of TKS5 was required for the capacity of TKS5 to rescue invadopodia formation (Figure 4D, E and G). In addition, quantification of a Mander's Overlap Coefficient as a measure of the colocalization between FGD1 and the overexpressed TKS5 constructs revealed that the integrity of TKS5 SH3#5 domain was similarly required for the colocalization of TKS5 and FGD1 in association with the collagen fibers (Figure 4E and F).

In reciprocal experiments, we investigated whether the overexpressed wild-type FGD1^{GFP} or a truncated mutant form of FGD1 with a deletion of the amino-terminal TKS5-binding domain Pro-rich domain (PRD domain) could rescue the loss of endogenous FGD1 for the formation of TKS5-positive invadopodia in MDA-MB-231 cells treated with the FGD1 siRNA. Overexpressed wild-type FGD1 rescued invadopodia formation in FGD1-depleted cells. The deletion of the PRD domain of FGD1 impaired invadopodia formation (Figure 4HI). All together, these new data indicate that TKS5 and FGD1 work in a linear pathway and that their interaction is required for invadopodia formation in MDA-MB-231 cells.

-Is everything done in one cell type? Could several, but not all, of the key findings be performed in a second or third cell type to show increased relevance?

As suggested by this referee, our revised manuscript now includes data obtained in three different cell lines: MDA-MB-231 and Hs578T, two triple negative breast cancer cell lines, and in the fibrosarcoma cell line, HT-1080. Our previous work highlighted a strong similarity in the MT1-MMP-, invadopodia-dependent invasion program in MDA-MB-231 and HT-1080 cells (Infante et al. Nat Comm 2018). New data included in this revised submission strengthen the relationship between these two cell lines as exemplified by the similitude in expression profiles of key invadopodia components. Especially, we observed that the functional high molecular weight isoform of TKS5 (known as TKS5 alpha), is highly expressed in both MDA-MB-231 and HT-1080 cells, while TKS4 levels are low or undetectable in these cell lines (Figure 3E). The situation is opposite in Hs578T cells, which express low-to-undetectable levels of TKS5 alpha and high levels of TKS4. These differences correlate with disparities in the morphology of the invadopodia structures in these cell lines, i.e. curvilinear in MDA-MB-231 and HT-1080 cells, in contrast with a straight morphology in Hs578T. Implications of these disparities are now discussed in the revised manuscript, including possible consequences for force production by TKS5- or TKS4-dependent invadopodia in light of our recent work relating curvature with invadopodia force generation (see Ferrari et al. Nat Comm 2019).

- This study provides insights into the type of PIP that is enriched at the invadopodial sites but does it provide functional insights into whether one PIP enhances recruitment and function of the proteins of interest to the degradation site or is this finding strictly observational?

In order to strengthen PIP localization data and in response to related comments and suggestions by Reviewer #1 and #3, we purchased Z-P045 anti-PI(4,5)P2 antibodies and Z-P034 anti-PI(3,4)P2 antibodies from Echelon Biosciences and stained MDA-MB-231 cells plated for 60 min on collagen fibers. Cells were counterstained for cortactin to label invadopodia. Unfortunately, in our hands, labeling looked unspecific and staining obtained with these two antibodies was not convincing. As we were not able to confirm and consolidate the phosphoinositide distribution data that we previously generated using genetically encoded PI(4,5)P2 and PI(3,4)P2 sensors, Tubby and TAPP1, respectively, we have decided to remove Tubby and TAPP1-based dataset in the revised manuscript.

- The second part of the study linking the protein complex to other effectors and lysosome distribution and matrix degradation make the story more diffuse and perhaps overdone. It's not clear if the players involved contribute directly to these processes or not. Pursuit of the suggestions above applied to these other

observations would help, or perhaps the overall story is too ambitious /complex and could be pared down.

We agree. In order to emphasize focus on the TKS5/FGD1/CDC42 module in the revised manuscript, all data related to the CDC42 effector protein, IQGAP1 have been deleted in the revised manuscript.

Reviewer #3:

In this paper, the authors set out to characterize the contribution and mechanism of TKS proteins in the formation of collagenolytic invadopodia in breast cancer cells. They have identified the interaction between TKS5 and FGD1 at the collagenolytic invadopodia and mapped down their interacting domains. They further unveiled this signaling pathway regulating collagenolytic invadopodia through CDC42 and IQGAP1. The experiments are very well-designed, and all the data strongly support the conclusions. With the improvement in quantification and certain extra experiments for validation, this paper should broaden the understanding of collagenolytic invadopodia formation.

We thank this Referee for her/his positive comments on our manuscript and for helpful suggestions.

The detail comments are as follows:

Whenever using the term of association/accumulation/colocalization in IF images, the authors are encouraged to quantify the colocalization in multiple cells. Showing just one cropped region of one cell without any quantitative analysis is not convincing.

Figure 1A-B, quantification for the colocalization of F-actin/cortactin with Col1 is needed for assessing their association.

The revised Figure 1AB, provides a new linescan-based correlation of the pixel fluorescence intensity of the two markers (cortactin and TKS5) along multiple collagen fiber-associated invadopodia. A detailed description of the analysis is provided in the Material and Methods section (Linescan-based correlation of pixel fluorescence intensity of invadopodia markers). Of note, Figure 1B has been moved to Supplementary Figure 1A and 1C in the revised manuscript.

The authors have concluded that the silencing of TKS5 strongly reduced the formation of F-actin-positive invadopodia (Figure 1B and Supplementary Figure 1A). Quantification of the number of invadopodia in the control and TKS5 KD cells is required for drawing this conclusion.

We agree that the quantification of invadopodia number can be of interest when using the gelatin substratum model, in which invadopodia form as homogeneous 0.1-0.5 μm diameter dotted shape structures that makes it easy to numerically quantify them. In addition, overall these structures are similar in their capacity to degrade the underlying gelatin substrate.

The situation is very different when using fibrillary type I collagen as a matrix construct like in the present study. Invadopodia forming in the association of the fibers can be very heterogenous in length, varying from less than 1 μm up to 20 μm . In addition, long curvilinear invadopodia can often be segmented into smaller structures with empty space between them making it too difficult

to score these long structures as either single or multiple invadopodia. Based on several years of observation, we would like to conclude that the collagenolytic activity of these invadopodia is linearly correlated with length, i.e. long invadopodia degrade more collagen than smaller ones. Yet a long proteolytically very active invadopodium would count as one structure, while small less active invadopodia would be counted as many structures. Instead, in all the analyses of invadopodia structures, we have preferred to quantify the area occupied by TKS5 signal over the total cell area as a measure of “TKS5-positive invadopodia” (Figure 4A, Figure 4I, Figure 5A, Figure 6C, Supplementary Figure 1D). The same rationale has been applied for the quantification of FGD1- (Figure 4G) or TKS4-positive invadopodia (Figure 6G).

Figure 1C, quantification for the colocalization of TKS5GFP with Col1-3/4C is needed for assessing their association.

Here the problem is that the Col1-3/4C signal is a cumulative one, i.e. cleaved collagen molecules accumulate over-time (usually 60-90 min in our experiments), while the TKS5 or cortactin signal is a snapshot of the structures existing at the time of fixation. Some Col1-3/4C signal may not be associated (anymore) with the TKS5 (or cortactin) signal as the corresponding invadopodia disassembled before fixation. We rather compare the overall Col1-3/4C signal generated during a given amount of time as a measure of invadopodia activity in different conditions.

- Figure 1G-H, it would be better to add a double KD of MT1-MMP and TKS5 to determine whether they have additive functions in cell invasion or not. This would be important to determine whether they are in one pathway as suggested by the model figure 7.

We would like to mention that we have reported that MT1-MMP knockdown prevents the formation of TKS5 invadopodia in a previous study (see Ferrari et al. Nat Comm 2019), thus we believe that MT1-MMP is required for invadopodia formation somehow upstream of TKS5 as discussed in the Discussion section and schematized in Figure 7D of the revised manuscript.

- The PI(4,5)P2 and PI(3,4)P2 biosensor data are not convincing. Figure 2A, first, the Col1 pattern is completely different from Figure 1A. Secondly, it is not clear what the big chunk of TubbyGFP(ideally should be PIP2) cropped by the authors is, which is used for demonstrating the association of PIP2 with TKS5 and Col1. It is more than 30 um in size. It may be just a cluster of TubbyGFP due to overexpression. Also, it is doubtful that the authors raised a statement that these data indicated homogenous

distribution of PI(4,5)P2 at the plasma membrane. As a secondary messenger, PI(4,5)P2 is also enriched in the signaling hotspot instead of being homogeneously distributed. Most importantly, TubbyGFP could not differentiate the real presence of PIP2 (TubbyGFP -PIP2 in the complex) or TubbyGFP alone. There is a PIP2 antibody for immunofluorescent staining (Z-P045, Echelon Biosciences). This experiment should be confirmed by immunofluorescent staining of PIP2. Beyond endogenous PI(4,5)P2 staining, the use of PI(3,4)P2 and PI(3,4,5)P3 antibodies (also available from Echelon Biosciences) to stain endogenous phosphoinositides at the invadopodia could further strengthen the authors conclusion.

As suggested by this Reviewer, in order to strengthen PI4,5P2 and PI3,4P2 distribution data and in response to related comments and suggestions by the other two Reviewers, we purchased the Z-P045 anti-PI(4,5)P2 antibodies and Z-P034 anti-PI(3,4)P2 antibodies from Echelon Biosciences and stained MDA-MB-231 cells plated for 60 min on collagen fibers. Cells were counterstained for cortactin to label invadopodia. Unfortunately, in our hands, labeling looked unspecific and staining obtained with these two antibodies was not convincing. As we were not able to confirm and consolidate the phosphoinositide distribution data that we initially generated using genetically encoded PI(4,5)P2 and PI(3,4)P2 sensors, Tubby and TAPP1, respectively, we have decided to remove Tubby and TAPP1-based dataset in the revised manuscript.

- Figure 2B, the uncropped images are required for showing how the cell mask staining works for the whole cell.

These data have been deleted in the revised manuscript.

- Figure 2C, poor quality image for the TAPP1GFP (PI(3,4)P2 biosensor). It is not clear if this is the real signal or noises cross-activated by other channels. Also, what the single white arrow-head in the green channel is pointing at? This may not be aligned properly, and the corresponding red channel is missing. Again, there is a statement of association of PI(3,4)P2 with TKS5/Col1 without quantification.

These data have been deleted in the revised manuscript.

- Figure 2D, no quantification of colocalization of INPP4B with cortactin/Col1 for staining accumulation. Figure 2E, no quantification of colocalization of p130cas with MT1-MMP/Col1 for staining colocalization.

As suggested by this referee, we have performed a new set of immunofluorescence analyses to address the distribution of endogenous SHIP2 and p130CAS in relation with cortactin. Similar analysis has been performed for overexpressed FLAG-tagged INNP4B and are reported in Figure 7EF. New images and quantification are reported in Figure 7A-C in the revised manuscript.

Figure 3C-D, the input for GST proteins and IB blot for the IP-ed GST proteins are missing. Also, it is important to have the quantification.

Figure 3 is Figure 2 in the revised manuscript. For Figure 2A-D, 5% of cell lysates have been loaded as a control (Input). We apologize for the missing information, which has been added in the legend of Figure 2.

Figure 4A-C, quantification is needed to conclude 'extensive co-localization'.

Figure 4A-C is now Figure 3A-C in the revised manuscript. Linescan-based correlation of the pixel fluorescence intensity of TKS5^{GFP} and FGD1 (Figure 3A), FGD1^{GFP} and TKS5 (Figure 3B) and FGD1 and Cortactin (Figure 3C) in multiple invadopodia in association with the collagen fibers is now provided in the corresponding figure panels.

Figure 4G, no quantification for stating accumulation/association.

These data have been deleted in the revised manuscript.

Figure 6B,E,D, no quantifications for stating association/co-localization.

Linescan-based correlation of the pixel fluorescence intensity of TKS4 and Cortactin in multiple invadopodia in association with the collagen fibers in MDA-MB-231 cells is now provided in Figure 6B.

Linescan-based correlation of the pixel fluorescence intensity of TKS4 and Cortactin (Figure 6D) and TKS5 and Cortactin (Figure 6E) in multiple invadopodia in association with the collagen fibers in Hs578T cells are provided in the corresponding Figure panels. In addition, we quantified the area covered by TKS4- or TKS5-positive structures over the whole cell surface and found that TKS4-positive invadopodia in Hs578T cells represented only ~40% of the area of TKS5-positive invadopodia in MDA-MB-231 cells.

Minor points:

1. *In vitro* and *in vivo* in the text should be italicized.

Done.

2. Figure 1G, no color code is labeled for the green staining.

Green labeling in Figure 1E of the revised manuscript corresponds to DAPI-stained nuclei. This information has been added in the legend of Figure 1.

3. There are two silQGAP1 #01 in Table S3. One should be silQGAP1 #03 according to Figure 5K, but this needs clarification.

IQGAP1 data have been deleted in the revised manuscript.

4. In the abstract "Here, using co-immunoprecipitation experiments, we identify a direct interaction between TKS5 and FGD1, which is required for the assembly and function of collagenolytic invadopodium." can be changed to "Here, using co-immunoprecipitation and *in vitro* pulldown experiment's, we identify a direct interaction ..."

Done.

Appended Figure for the Referees

Association of CDC42 and CDC42 V12 with collagenolytic invadopodia. (A) MDA-MB-231 cells transfected with $mChTKS5$ (red) and GFP^{CDC42} WT were seeded on type I collagen (blue) for 60min. Scale bar, 10 μm ; zoom-in of boxed region, scale bar, 5 μm . (B,C) The gallery shows non-consecutive frames from a representative movie of a MDA-MB-231 cell expressing GFP-CDC42 WT (B, green) or GFP-CDC42 V12 (C, green) on a type I collagen fibrillary layer (magenta). Time is in min. Scale bar, 10 μm . (D) Images of the type I collagen fibrillary layer at 0 min and 60 min underneath GFP-CDC42 WT (left) and GFP-CDC42 V12-expressing cells (right).

May 22, 2020

RE: JCB Manuscript #201910132R

Dr. Philippe Chavier
Institut Curie
Institut Curie - Section Recherche CNRS UMR144 26 rue d'Ulm
Paris 75248
France

Dear Dr. Chavier,

Thank you for submitting your revised manuscript entitled "Intersection of TKS5 and FGD1/CDC42 signaling cascades directs the formation of invadopodia". You will see that the reviewers are largely positive, but one referee shares some final points that require your attention. While we will not absolutely require new data to address these points, like the referee, we are still interested in the suggested experiments with recombinant proteins. If they are possible at all when labs are open, we would encourage you to move forward with these experiments. However, we remain interested in the study whether or not you can add data. We would be happy to further consider your paper for publication in JCB pending final revisions necessary to meet our formatting guidelines (see details below) and pending your best efforts to address the final two remaining concerns of the peer-reviewing process. Please do not hesitate to contact us with any questions.

1) eTOC summary: A 40-word summary that describes the context and significance of the findings for a general readership should be included on the title page. The statement should be written in the present tense and refer to the work in the third person.

- Please include a summary statement on the title page of the resubmission. ***It should start with "First author name(s) et al..." to match our preferred style.***

2) Figure formatting: Scale bars must be present on all microscopy images, including inset magnifications. Please add scale bars to 1E, &ABC (bottom images)

3) Statistical analysis: Error bars on graphic representations of numerical data must be clearly described in the figure legend. The number of independent data points (n) represented in a graph must be indicated in the legend. Statistical methods should be explained in full in the materials and methods. For figures presenting pooled data the statistical measure should be defined in the figure legends.

Please indicate n/sample size/how many experiments the data are representative of: 4ABGI, fig 5, 6FG, S1DE

4) Materials and methods: Should be comprehensive and not simply reference a previous publication for details on how an experiment was performed. Please provide full descriptions in the text for readers who may not have access to referenced manuscripts.

- please be sure to include sequences for all siRNA oligos, including negative controls if available to

you.

- Microscope image acquisition: The following information must be provided about the acquisition and processing of images:

- a. Make and model of microscope
- b. Type, magnification, and numerical aperture of the objective lenses
- c. Temperature
- d. imaging medium
- e. Fluorochromes
- f. Camera make and model
- g. Acquisition software
- h. Any software used for image processing subsequent to data acquisition. Please include details and types of operations involved (e.g., type of deconvolution, 3D reconstitutions, surface or volume rendering, gamma adjustments, etc.).

5) A summary paragraph of all supplemental material should appear at the end of the Materials and methods section.

6) Author contributions: A separate author contribution section is required following the Acknowledgments in all research manuscripts. All authors should be mentioned and designated by their full names. We encourage use of the CRediT nomenclature.

A. MANUSCRIPT ORGANIZATION AND FORMATTING:

Full guidelines are available on our Instructions for Authors page, <http://jcb.rupress.org/submission-guidelines#revised>. **Submission of a paper that does not conform to JCB guidelines will delay the acceptance of your manuscript.**

B. FINAL FILES:

-- High-resolution figure and video files: See our detailed guidelines for preparing your production-ready images, <http://jcb.rupress.org/fig-vid-guidelines>.

**The license to publish form must be signed before your manuscript can be sent to production. A link to the electronic license to publish form will be sent to the corresponding author only. Please

take a moment to check your funder requirements before choosing the appropriate license.**

Thank you for this interesting contribution, we look forward to publishing your paper in the Journal of Cell Biology.

Sincerely,

Kenneth Yamada, MD, PhD
Editor, Journal of Cell Biology

Melina Casadio, PhD
Senior Scientific Editor, Journal of Cell Biology

Reviewer #1 (Comments to the Authors (Required)):

The authors performed specific additional experiments to strengthen the relevance of the purported novel linear TSK5-FGD1-CDC42 pathways in the control of invadopodia. They also clarify the additional points raised.

Reviewer #2 (Comments to the Authors (Required)):

The manuscript has been improved somewhat and is more focused. A few relevant issues remain that could be fixed if need be but perhaps not essential.

Original Reviewer query--It is unclear how this interaction was first identified and by whom. Was it first reported by Hess and colleagues using a Mann-like screen and this current group performed the needed characterization and cell biological analysis? So perhaps this interaction was already known?

Author response--We are aware about one study from Matthias Mann's laboratory describing a global human interactome study, in which an interaction between TKS5 and FGD1 has been reported (Hein et al., Cell. 163:712-723, 2015) but was not further validated. This study is cited in our manuscript.

Continued Reviewer concern-I raised this point as my understanding from the Hein study is that THEY identified the novel interaction but did not choose to pursue it further. Is it not disingenuous for these authors to claim that they identified a novel interaction 4 years later as it currently reads in the abstract and the introduction? Simply citing the past study out of its true context is besides the point and doesn't address the real concern. Would it not be more genuine and perhaps

accurate to describe the Hein observation right up front and state that, "because of the potential importance of this putative interaction identified by others, this current study defines the functional relevance in the context of stromal remodeling by tumor cells ?" Perhaps this approach reduces the impact of the current study a bit but to this reviewer that is the honest chronology of events.

Original Reviewer Query--Are the reported interactions direct or indirect? It appears that all the GST or GFP trap pulldowns were conducted in cell lysates so is there an adaptor mediating these interactions?

Author Response--Although we agree that the issue as to whether the interaction between FGD1 and TKS5 is direct or not is of interest, biochemical assessment of this interaction would require recombinant proteins that were not easily available to us.

Continued Reviewer concern---The authors provide some new insights into the interactions of the players but have not defined a direct interaction which is a bit surprising as the story is heavily reliant on protein-protein connections of a new "network." Not sure why obtaining recombinant proteins to test this was deemed insurmountable.

Reviewer #3 (Comments to the Authors (Required)):

This is now a solid manuscript. The authors have responded largely to the reviewer's comments and this has improved the manuscript. There remain minor issues but these should not slow publication at this time.

Journal of Cell Biology

May 26th, 2020

Re: JCB manuscript #201910132R – Final Draft

Dear Drs. Yamada and Casadio,

We would like to thank you for your support and continued interest in our work and for the possibility to publish our article in the Journal of Cell Biology.

Regarding the Reviewer #2's comment concerning the previous identification of the TKS5/FGD1 interaction. As we report in the present manuscript, we identified FGD1 among several proteins that were immunoprecipitated together with TKS5^{GFP} in MDA-MB-231 cells cultured in the presence of fibrillary type I collagen. This interaction was already known from the outstanding work published by Hein et al. (Cell 2015) describing a global human interactome comprising ~32,000 different interactions, including the FGD1/TKS5 interaction. Based on Hein et al. study and on previous work from several labs including Elisabeth Génot's lab, one of our coauthors, identifying FGD1 as a CDC42 GEF involved in podosome and invadopodia function we have pursued the characterization of what seemed to be a promising avenue of research. Yet, it was certainly never our intention to minimize the scope of the initial study by Hein and colleagues. To further clarify this issue, we have rephrased the Introduction and Discussion section as we refer to the work by Hein et al. as suggested by Referee #2.

Introduction: 'Of note, these findings confirmed a previously identified TKS5/FGD1 interaction from a global human interactome study (Hein et al., 2015).'

Discussion: 'The TKS5/FGD1 interaction, which was initially reported in a global human interactome study (Hein et al., 2015), is validated in the present work in the context of stromal remodeling by tumor cells.'

Concerning Reviewer #2's concern about a lack of evidence for a direct interaction between TKS5 and FGD1. We would like to emphasize that we used several approaches to narrow down the interaction domains in both proteins and have found that the amino-terminal proline-rich domain of FGD1 and the fifth carboxy-terminal SH3 domain of TKS5 are responsible for the FGD1/TKS5 interaction. The alternative approaches we deployed to map the interaction domains in FGD1 and TKS5 have kept us away from a more definitive demonstration of a direct TKS5-FGD1 interaction. Because of COVID-19-related disruption of our activities, we feel that the final demonstration of a direct interaction, although not insurmountable, would certainly delay very significantly this submission. At this stage, we would also like to emphasize the fact that, although our data do not rule out the possibility of another protein bridging FGD1 and TKS5 in the complex, it is very conceivable that this interaction is indeed a direct one as SH3 domains typically interact via binding to proline-rich peptides in their respective binding partner.

Finally, as requested eTOC summary and author contributions have been added, scale bars are present in all figures and we added missing 'n/sample size/how many experiments' information in the legend of 4ABGI, fig 5, 6FG, S1DE.

We hope that you will now find this work suitable for publication in *Journal of Cell Biology*.

Sincerely,

Philippe Chavrier, PhD